# ODE Analysis of Stochastic Gradient Methods with Optimism and Anchoring for Minimax Problems and GANs

## Abstract

Despite remarkable empirical success, the training dynamics of generative adversarial networks (GAN), which involves solving a minimax game using stochastic gradients, is still poorly understood. In this work, we analyze last-iterate convergence of simultaneous gradient descent (simGD) and its variants under the assumption of convex-concavity, guided by a continuous-time analysis with differential equations. First, we show that simGD, as is, converges with stochastic sub-gradients under strict convexity in the primal variable. Second, we generalize optimistic simGD to accommodate an optimism rate separate from the learning rate and show its convergence with full gradients. Finally, we present anchored simGD, a new method, and show convergence with stochastic subgradients.

## 1 Introduction

Training of generative adversarial networks (GAN) (Goodfellow et al., 2014), solving a minimax game using stochastic gradients, is known to be difficult. Despite the remarkable empirical success of GANs, further understanding the global training dynamics empirically and theoretically is considered a major open problem (Goodfellow, 2016; Radford et al., 2016; Metz et al., 2017; Mescheder et al., 2018; Odena, 2019).

The **local** training dynamics of GANs is understood reasonably well. Several works have analyzed convergence assuming the loss functions have linear gradients and assuming the training uses **full** (deterministic) gradients. Although the linear gradient assumption is reasonable for local analysis (even though the loss functions may not be continuously differentiable due to ReLU activation functions) such results say very little about global convergence. Although the full gradient assumption is reasonable when the learning rate is small, such results say very little about how the randomness affects the training.

This work investigates **global** convergence of simultaneous gradient descent (simGD) and its variants for zero-sum games with a convex-concave cost using using **stochastic sub**gradients. We specifically study convergence of the last iterates as opposed to the averaged iterates.

**Organization.** Section 2 presents convergence of simGD with stochastic subgradients under strict convexity in the primal variable. The goal is to establish a minimal sufficient condition of global convergence for simGD without modifications. Section 3 presents a generalization of *optimistic* simGD (Daskalakis et al., 2018), which allows an optimism rate separate from the learning rate. We prove the generalized optimistic simGD using full gradients converges, and experimentally demonstrate that the optimism rate must be tuned separately from the learning rate when using stochastic gradients. However, it is unclear whether optimistic simGD is theoretically compatible with stochastic gradients. Section 4 presents *anchored* simGD, a new method, and presents its convergence with stochastic subgradients. Anchoring represents what we consider to be the strongest contribution of this work. The presentation and analyses of Sections 2, 3, and 4 are guided by continuous-time first-order ordinary differential equations (ODE). In particular, we interpret optimism and anchoring as discretizations of certain regularized dynamics. Section 5 experimentally demonstrates the benefit of optimism and anchoring for training GANs in some setups.

**Prior work.** There are several independent directions for improving the training of GANs such as designing better architectures, choosing good loss functions, or adding appropriate regularizers (Radford et al., 2016; Arjovsky et al., 2017; Sønderby et al., 2017; Arjovsky & Bottou, 2017; Gulrajani et al., 2017; Wei et al., 2018; Roth et al., 2017; Mescheder et al., 2018; 2017; Miyato et al., 2018). In this work, we accept these factors as a given and focus on how to train (optimize) the model effectively.

*Optimism* is a simple modification to remedy the cycling behavior of simGD, which can occur even under the bilinear convex-concave setup (Daskalakis et al., 2018; Daskalakis & Panageas, 2018; 2019; Mertikopoulos et al., 2019; Gidel et al., 2019a; Liang & Stokes, 2019; Mokhtari et al., 2019; Peng et al., 2019). These prior work assume the gradients are linear and use full gradients. Although the recent name 'optimism' originates from its use in online optimization (Chiang et al., 2012; Rakhlin & Sridharan, 2013a;b; Syrgkanis et al., 2015), the idea dates back to Popov's work in the 1980s (Popov, 1980) and has been studied independently in the mathematical programming community (Malitsky & Semenov, 2014; Malitsky, 2015; Malitsky & Tam, 2018; Malitsky, 2019; Csetnek et al., 2019).

We note that there are other mechanisms similar to optimism and anchoring such as "prediction" (Yadav et al., 2018), "negative momentum" (Gidel et al., 2019b), and "extragradient" (Korpelevich, 1976; Tseng, 2000; Chavdarova et al., 2019). In this work, we focus on optimism and anchoring.

Classical literature analyze convergence of the Polyak-averaged iterates (which assigns less weight to newer iterates) when solving convex-concave saddle point problems using stochastic subgradients (Bruck, 1977; Nemirovski & Yudin, 1978; Nemirovski et al., 2009; Juditsky et al., 2011; Gidel et al., 2019a). For GANs, however, last iterates or exponentially averaged iterates (Yazıcı et al., 2019) (which assigns more weight to newer iterates) are used in practice. Therefore, the classical work with Polyak averaging do not fully explain the empirical success of GANs.

We point out that we are not the first to utilize classical techniques for analyzing the training of GANs. In particular, the stochastic approximation technique (Heusel et al., 2017; Duchi & Ruan, 2018), control theoretic techniques (Heusel et al., 2017; Nagarajan & Kolter, 2017), ideas from variational inequalities and monotone operator theory (Gemp & Mahadevan, 2018; Gidel et al., 2019a), and continuous-time ODE analysis (Heusel et al., 2017; Csetnek et al., 2019) have been utilized for analyzing GANs.

## 2 STOCHASTIC SIMULTANEOUS SUBGRADIENT DESCENT

Consider the cost function $L : \mathbb{R}^m \times \mathbb{R}^n \to \mathbb{R}$ and the minimax game $\min_x \max_u L(x, u)$. We say $(x_\star, u_\star) \in \mathbb{R}^m \times \mathbb{R}^n$ is a solution to the minimax game or a *saddle point* of $L$ if

$$L(x_\star, u) \leq L(x_\star, u_\star) \leq L(x, u_\star), \qquad \forall x \in \mathbb{R}^m, \, u \in \mathbb{R}^n.$$

We assume

$$L \text{ is convex-concave and has a saddle point.} \tag{A0}$$

By convex-concave, we mean $L(x, u)$ is a convex function in $x$ for fixed $u$ and a concave function in $u$ for fixed $x$. Define

$$G(x, u) = \begin{bmatrix} \partial_x L(x, u) \\ \partial_u (-L(x, u)) \end{bmatrix},$$

where $\partial_x$ and $\partial_u$ respectively denote the subdifferential with respect to $x$ and $u$. For simplicity, write $z = (x, u) \in \mathbb{R}^{m+n}$ and $G(z) = G(x, u)$. Note that $0 \in G(z)$ if and only if $z$ is a saddle point. Since $L$ is convex-concave, the operator $G$ is monotone (Rockafellar, 1970):

$$(g_1 - g_2)^T (z_1 - z_2) \geq 0 \qquad \forall g_1 \in G(z_1), \, g_2 \in G(z_2), \, z_1, z_2 \in \mathbb{R}^{m+n}. \tag{1}$$

Let $g(z; \omega)$ be a stochastic subgradient oracle, i.e., $\mathbb{E}_\omega g(z; \omega) \in G(z)$ for all $z \in \mathbb{R}^{m+n}$, where $\omega$ is a random variable. Consider *Simultaneous Stochastic Sub-Gradient Descent*

$$z_{k+1} = z_k - \alpha_k g(z_k; \omega_k) \tag{SSSGD}$$

for $k = 0, 1, \ldots$, where $z_0 \in \mathbb{R}^{m+n}$ is a starting point, $\alpha_0, \alpha_1, \ldots$ are positive learning rates, and $\omega_0, \omega_1, \ldots$ are IID random variables. (We read SSSGD as "triple-SGD".) In this section, we provide convergence of SSSGD when $L(x, u)$ is strictly convex in $x$.

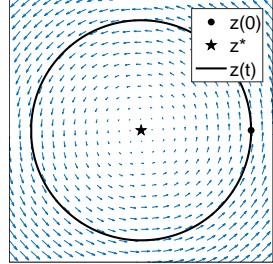 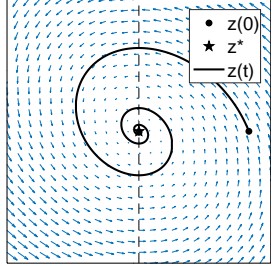

Figure 1: $z(t)$ with $\dot{z}(t) = -G(z(t))$. (Left) $L(x, u) = xu$. All points satisfy $G(z)^T(z - z_\star) = 0$ so $\|z(t) - z_\star\|$ does not decrease and $z(t)$ forms a cycle. (Right) $L(x, u) = 0.2x^2 + xu$. The dashed line denotes where $G(z)^T(z - z_\star) = 0$, but it is visually clear that $z_\star = 0$ is the only cluster point.

## 2.1 CONTINUOUS-TIME ILLUSTRATION

To understand the asymptotic dynamics of the stochastic discrete-time system, we consider a corresponding deterministic continuous-time system. For simplicity, assume $G$ is single-valued and smooth. Consider

$$\dot{z}(t) = -g(t), \qquad g(t) = G(z(t))$$

with an initial value $z(0) = z_0$. (We introduce $g(t)$ for notational simplicity.) Let $z_\star$ be a saddle point, i.e., $G(z_\star) = 0$. Then $z(t)$ does not move away from $z_\star$:

$$\frac{d}{dt}\frac{1}{2}\|z(t) - z_\star\|^2 = -g(t)^T(z(t) - z_\star) \leq 0,$$

where we used (1). However, there is no mechanism forcing $z(t)$ to converge to a solution.

Consider the two examples $L_0(x, u) = xu$ and $L_\rho(x, u) = (\rho/2)x^2 + xu$ with

$$G_0(x, u) = \begin{bmatrix} 0 & 1 \\ -1 & 0 \end{bmatrix} \begin{bmatrix} x \\ u \end{bmatrix}, \qquad G_\rho(x, u) = \begin{bmatrix} \rho & 1 \\ -1 & 0 \end{bmatrix} \begin{bmatrix} x \\ u \end{bmatrix} \tag{2}$$

where $x \in \mathbb{R}$ and $u \in \mathbb{R}$ and $\rho > 0$. Note that $L_0$ is the canonical counter example that also arises as the Dirac-GAN (Mescheder et al., 2018). See Figure 1.

The classical LaSalle–Krasnovskii invariance principle (Krasovskii, 1959; LaSalle, 1960) states (paraphrased) if $z_\infty$ is a cluster point of $z(t)$, then the dynamics starting at $z_\infty$ will have a constant distance to $z_\star$. On the left of Figure 1, we can see $\|z(t) - z_\star\|^2$ is constant as $\frac{d}{dt}\frac{1}{2}\|z(t) - z_\star\|^2 = 0$ for all $t$. On the right of Figure 1, we can see that although $\frac{d}{dt}\frac{1}{2}\|z(t) - z_\star\|^2 = 0$ when $z(t) = (0, u)$ for $u \neq 0$ (the dotted line) this 0 derivative is temporary as $z(t)$ will soon move past the dotted line. Therefore, $z(t)$ can maintain a constant constant distance to $z_\star$ only if it starts at 0, and 0 is the only cluster point of $z(t)$.

## 2.2 DISCRETE-TIME CONVERGENCE ANALYSIS

Consider the further assumptions

$$\sum_{k=0}^{\infty} \alpha_k = \infty, \qquad \sum_{k=0}^{\infty} \alpha_k^2 < \infty \tag{A1}$$

$$\mathbb{E}_{\omega_1, \omega_2}\|g(z_1; \omega_1) - g(z_2; \omega_2)\|^2 \leq R_1^2\|z_1 - z_2\|^2 + R_2^2 \quad \forall\, z_1, z_2 \in \mathbb{R}^{m+n}, \tag{A2}$$

where $\omega_1$ and $\omega_2$ are independent random variables and $R_1 \geq 0$ and $R_2 \geq 0$. These assumptions are standard in the sense that analogous assumptions are used in convex minimization to establish almost sure convergence of stochastic gradient descent.

**Theorem 1.** *Assume* (A0)*,* (A1)*, and* (A2)*. Furthermore, assume $L(x, u)$ is strictly convex in $x$ for all $u$. Then SSSGD converges in the sense of $z_k \xrightarrow{a.s.} z_\star$ where $z_\star$ is a saddle point of $L$.*

We can alternatively assume $L(x, u)$ is strictly concave in $u$ for all $x$ and obtain the same result.

The proof uses the stochastic approximation technique of (Duchi & Ruan, 2018). We show that the discrete-time process converges (in an appropriate topology) to a continuous-time trajectory satisfying a differential inclusion and use the LaSalle–Krasnovskii invariance principle to argue that cluster points are solutions.

**Related prior work.** Theorem 3.1 of (Mertikopoulos et al., 2019) considers the more general mirror descent setup and proves convergence under the assumption of "strict coherence", which is analogous to the stronger assumption of strict convex-concavity in both $x$ and $u$.

## 3 SIMULTANEOUS GD WITH OPTIMISM

Consider the setup where $L$ is continuously differentiable and we access full (deterministic) gradients

$$G(x, u) = \begin{bmatrix} \nabla_x L(x, u) \\ -\nabla_u L(x, u) \end{bmatrix}.$$

Consider *Optimistic Simultaneous Gradient Descent*

$$z_{k+1} = z_k - \alpha G(z_k) - \beta(G(z_k) - G(z_{k-1})) \tag{SimGD-O}$$

for $k \geq 0$, where $z_0 \in \mathbb{R}^{m+n}$ is a starting point, $z_{-1} = z_0$, $\alpha > 0$ is learning rate, and $\beta > 0$ is the *optimism rate*. Optimism is a modification to simGD that remedies the cycling behavior; for the bilinear example $L_0$ of (2), simGD (case $\beta = 0$) diverges while SimGD-O with appropriate $\beta > 0$ converges. In this section, we provide a continuous-time interpretation of SimGD-O as a regularized dynamics and provide convergence for the deterministic setup.

### 3.1 CONTINUOUS-TIME ILLUSTRATION

Consider the continuous-time dynamics

$$\dot{z}(t) = -\alpha g(t) - \beta \dot{g}(t), \qquad g(t) = G(z(t)).$$

The discretization $\dot{z}(t) \approx z_{k+1} - z_k$ and $\dot{g}(t) \approx G(z_k) - G(z_{k-1})$ yields SimGD-O. We discuss how this system arises as a certain regularized dynamics and derive the convergence rate

$$\|g(t)\|^2 \leq \mathcal{O}(1/t).$$

**Regularized gradient mapping.** The Moreau–Yosida (Moreau, 1965; Yosida, 1948) regularization of $G$ with parameter $\beta > 0$ is

$$G_\beta = \beta^{-1}(I - (I + \beta G)^{-1}).$$

To clarify, $I : \mathbb{R}^{m+n} \to \mathbb{R}^{m+n}$ is the identity mapping and $(I + \beta G)^{-1}$ is the inverse (as a function) of $I + \beta G$, which is well-defined by Minty's theorem (Minty, 1962). It is straightforward to verify that $G_\beta(z) = 0$ if and only if $G(z) = 0$, i.e., $G_\beta$ and $G$ share the same equilibrium points. For small $\beta$, we can think of $G_\beta$ as an approximation $G$ that is better-behaved. Specifically, $G$ is merely monotone (satisfies (1)), but $G_\beta$ is furthermore $\beta$-cocoercive, i.e.,

$$(G_\beta(z_1) - G_\beta(z_2))^T(z_1 - z_2) \geq \beta \|G_\beta(z_1) - G_\beta(z_2)\|^2 \qquad \forall z_1, z_2 \in \mathbb{R}^{m+n}. \tag{3}$$

**Regularized dynamics.** Consider the regularized dynamics

$$\dot{\zeta}(t) = -\alpha G_\beta(\zeta(t)).$$

Reparameterize the dynamics $\dot{\zeta}(t) = -\alpha G_\beta(\zeta(t))$ with $z(t) = (I + \beta G)^{-1}(\zeta(t))$ and $g(t) = G(z(t))$ to get $\zeta(t) = z(t) + \beta g(t)$ and

$$\dot{z}(t) + \beta \dot{g}(t) = \dot{\zeta}(t) = -\frac{\alpha}{\beta}(\zeta(t) - z(t)) = -\alpha g(t).$$

This gives us $\dot{z}(t) = -\alpha g(t) - \beta \dot{g}(t)$.

**Rate of convergence.** We now derive a rate of convergence. Let $z_\star$ satisfy $G(z_\star) = 0$ (and therefore $G_\beta(z_\star) = 0$). Then

$$\frac{d}{dt}\frac{1}{2}\|\zeta(t) - z_\star\|^2 = (\zeta(t) - z_\star)^T \dot\zeta(t) = -\alpha(\zeta(t) - z_\star)^T G_\beta(\zeta(t))$$
$$\leq -\alpha\beta\|G_\beta(\zeta(t))\|^2,$$

where we use cocoercivity, (3). This translates to

$$\frac{d}{dt}\frac{1}{2}\|z(t) + \beta g(t) - z_\star\|^2 \leq -\alpha\beta\|g(t)\|^2. \tag{4}$$

The quantity $\|g(t)\|^2$ is nonincreasing since

$$\frac{d}{dt}\frac{1}{2}\|g(t)\|^2 = -\frac{1}{\alpha}\dot\zeta(t)^T \dot g(t) = -\frac{1}{\alpha}\lim_{h\to 0}\frac{1}{h^2}(\zeta(t+h) - \zeta(t))^T(G_\beta(\zeta(t+h)) - G_\beta(\zeta(t)))$$
$$\leq -\frac{\beta}{\alpha}\lim_{h\to 0}\frac{1}{h^2}\|G_\beta(\zeta(t+h)) - G_\beta(\zeta(t))\| = -\frac{\beta}{\alpha}\|\dot g(t)\|^2 \leq 0,$$

where we use cocoercivity, (3). Finally, integrating (4) on both sides gives us

$$\frac{1}{2}\|z(t) + \beta g(t) - z_\star\|^2 - \frac{1}{2}\|z(0) + \beta g(0) - z_\star\|^2 \leq -\alpha\beta\int_0^t \|g(s)\|^2\, ds \leq -\alpha\beta t\|g(t)\|^2$$

$$\|g(t)\|^2 \leq \frac{1}{2\alpha\beta t}\|z(0) + \beta g(0) - z_\star\|^2.$$

**Related prior work.** The use of the Moreau–Yoshida regularization for the continuous-time analysis was inspired by Attouch et al. (Attouch et al., 2002; Attouch & Peypouquet, 2019) who first used the Moreau–Yosida regularization in continuous-time dynamics and Csetnek et al. (2019) who interpreted a forward-backward-forward-type method as a discretization of continuous-time dynamics with the Douglas–Rachford operator. Daskalakis et al. (2018) interprets optimism as augmenting "follow the regularized leader" with the (optimistic) prediction that the next gradient will be the same as the current gradient in online learning setup. Peng et al. (2019) interprets optimism as "centripetal acceleration" but does not provide a formal analysis with differential equations.

### 3.2 DISCRETE-TIME CONVERGENECE ANALYSIS

The discrete-time method SimGD-O converges under the assumption

$$L \text{ is differentiable and } \nabla L \text{ is } R\text{-Lipschitz continuous.} \tag{A3}$$

**Theorem 2.** *Assume* (A0) *and* (A3). *If* $0 < \alpha < 2\beta(1 - 2\beta R)$, *then SimGD-O converges in the sense of*

$$\min_{i=0,\ldots,k}\|G(z_k)\|^2 \leq \frac{2 + 2\beta^2 R^2}{\alpha(2\beta - \alpha - 4\beta^2 R)k}\|z_0 + \beta G(z_0) - z_\star\|^2.$$

*Furthermore,* $z_k \to z_\star$, *where* $z_\star$ *is a saddle point of* $L$.

The proof can be considered a discretization of the continuous-time analysis. We further discuss the similarities and differences between the continuous and discrete analyses in Section A.

**Corollary 1.** *In the setup of Theorem 2, the choice* $\alpha = 1/(8R)$ *and* $\beta = 2\alpha$ *yields*

$$\min_{i=0,\ldots,k}\|G(z_k)\|^2 \leq \frac{136R^2}{k}\|z_0 + \beta G(z_0) - z_\star\|^2 \leq \frac{289R^2}{k}\|z_0 - z_\star\|^2.$$

**Related prior work.** Peng et al. (2019) show convergence of simGD-O for $\alpha \neq \beta$ and bilinear $L$. Malitsky & Tam (2018) and Csetnek et al. (2019) show convergence of simGD-O for $\alpha = \beta$ and convex-concave $L$. Theorem 2 establishes convergence for $\alpha \neq \beta$ and convex-concave $L$ and presents an explicit rate.

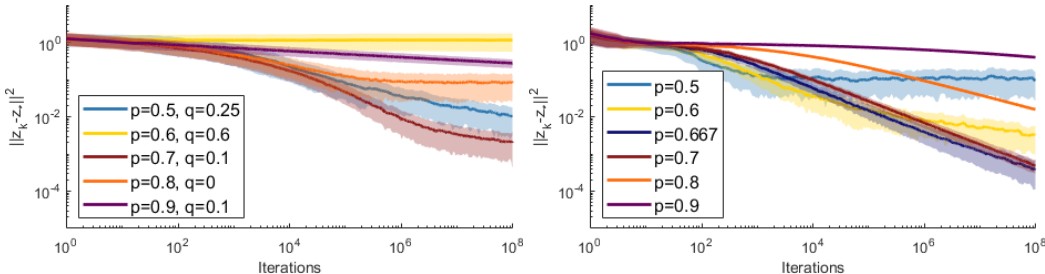

Figure 2: Plot of $\|z_k - z_\star\|^2$ vs. iteration count for simGD-OS (left) and SSSGD-A (right) with $\alpha_k = 1/k^p$ and $\beta_k = 1/k^q$. We use $L_0$ of (2) and Gaussian random noise. The shaded region denotes $\pm$ standard error. For simGD-OS, we see that neither $q = 0$ nor $q = p$ leads to convergence. Rather, $q$ must satisfy $0 < q < p$ so that the learning rate diminishes faster than the optimism rate.

### 3.3 DIFFICULTY WITH STOCHASTIC GRADIENTS

Training in machine learning usually relies on stochastic gradients, rather than full gradients. We can consider a stochastic variation of SimGD-O:

$$z_{k+1} = z_k - \alpha_k g(z_k; \omega_k) - \beta_k(g(z_k; \omega_k) - g(z_{k-1}; \omega_{k-1})) \qquad \text{(SimGD-OS)}$$

with learning rate $\alpha_k$ and optimism rate $\beta_k$.

Figure 2 presents experiments of SimGD-OS on a simple bilinear problem. The choice $\beta_k = \alpha_k$ where $\alpha_k \to 0$ does not lead to convergence. Discretizing $\dot{z}(t) = -\alpha g(t) - \beta \dot{g}(t)$ with a diminishing step $h_k$ leads to the choice $\alpha_k = \alpha h_k$ and $\beta_k = \beta$, but this choice does not lead to convergence either. Rather, it is necessary to tune $\alpha_k$ and $\beta_k$ separately as in Theorem 2 to obtain convergence and dynamics appear to be sensitive to the choice of $\alpha_k$ and $\beta_k$. In particular, both $\alpha_k$ and $\beta_k$ must diminish and $\alpha_k$ must diminish faster than $\beta_k$. One explanation of this difficulty is that the finite difference approximation $\alpha_k^{-1}(g(z_k; \omega_k) - g(z_{k-1}; \omega_{k-1})) \approx \dot{g}(t)$ is unreliable when using stochastic gradients.

Whether the observed convergence holds generally in the nonlinear convex-concave setup and whether optimism is compatible with subgradients is unclear. This motivates anchoring of the following section which is provably compatible with stochastic subgradients.

**Related prior work.** Gidel et al. (2019a) show averaged iterates of SimGD-OS converge if iterates are projected onto a compact set. Mertikopoulos et al. (2019) show almost sure convergence of SimGD-OS under strict convex-concavity (and more generally under "strict coherence"). However, such analyses do not provide a compelling reason to use optimism since SimGD without optimism already converges under these setups.

## 4 SIMULTANEOUS GD WITH ANCHORING

Consider setup of Section 3. We propose *Anchored Simultaneous Gradient Descent*

$$z_{k+1} = z_k - \frac{1-p}{(k+1)^p} G(z_k) + \frac{(1-p)\gamma}{k+1}(z_0 - z_k) \qquad \text{(SimGD-A)}$$

for $k \geq 0$, where $z_0 \in \mathbb{R}^{m+n}$ is a starting point, $p \in (1/2, 1)$, and $\gamma > 0$ is the *anchor rate*. In this section, we provide a continuous-time illustration of SimGD-A and provide convergence for both the deterministic and stochastic setups.

### 4.1 CONTINUOUS-TIME ILLUSTRATION

Consider the continuous-time dynamics

$$\dot{z}(t) = -g(t) + \frac{\gamma}{t}(z_0 - z(t)), \qquad g(t) = G(z(t)).$$

for $t \geq 0$, where $\gamma \geq 1$ and $z(0) = z_0$. We will derive the convergence rate

$$\|g(t)\|^2 \leq \mathcal{O}(1/t^2).$$

Discretizing the continuous-time ODE with diminishing steps $(1-p)/(k+1)^p$ leads to SimGD-A.

**Rate of convergence.**  First note

$$0 \leq \frac{1}{h^2} \langle z(t+h) - z(t), g(t+h) - g(t) \rangle \to \langle \dot{z}(t), \dot{g}(t) \rangle \qquad \text{as } h \to 0.$$

Using this, we have

$$\frac{d}{dt} \frac{1}{2} \|\dot{z}(t)\|^2 = - \left\langle \dot{z}(t), \dot{g}(t) + \frac{\gamma}{t}\dot{z}(t) + \frac{\gamma}{t^2}(z_0 - z(t)) \right\rangle$$

$$= -\langle \dot{z}(t), \dot{g}(t) \rangle - \frac{\gamma}{t}\|\dot{z}(t)\|^2 + \frac{\gamma}{t^2}\langle z(t) - z_0, \dot{z} \rangle$$

$$\leq -\frac{\gamma}{t}\|\dot{z}(t)\|^2 + \frac{\gamma}{t^2}\langle z(t) - z_0, \dot{z} \rangle.$$

Using $\gamma \geq 1$, we have

$$\frac{d}{dt} \frac{1}{2} \|\dot{z}(t)\|^2 + \frac{1}{t}\|\dot{z}(t)\|^2 \leq \frac{\gamma}{t^2}\langle z(t) - z_0, \dot{z} \rangle.$$

Multiplying by $t^2$ and integrating both sides gives us

$$\frac{t^2}{2}\|\dot{z}(t)\|^2 \leq \frac{\gamma}{2}\|z(t) - z_0\|^2.$$

Reorganizing, we get

$$\frac{t^2}{2}\|g(t)\|^2 - \gamma t \langle g(t), z_0 - z(t) \rangle + \frac{\gamma^2}{2}\|z(t) - z_0\|^2 \leq \frac{\gamma}{2}\|z(t) - z_0\|^2$$

Using $\gamma \geq 1$, the monotonicity inequality, and Young's inequality, we get

$$\|g(t)\|^2 \leq \frac{2\gamma}{t}\langle g(t), z_0 - z(t) \rangle \leq \frac{2\gamma}{t}\langle g(t), z_0 - z_\star \rangle \leq \frac{1}{2}\|g(t)\|^2 + \frac{2\gamma^2}{t^2}\|z_0 - z_\star\|^2$$

and conclude

$$\|g(t)\|^2 \leq \frac{4\gamma^2}{t^2}\|z_0 - z_\star\|^2.$$

Interestingly, anchoring leads to a faster rate $\mathcal{O}(1/t^2)$ compared to the rate $\mathcal{O}(1/t)$ of optimism in continuous time. The discretized method, however, is not faster than $\mathcal{O}(1/k)$. We further discuss this difference in Section A.

**Related prior work.**  Anchoring was inspired by Halpern's method (Halpern, 1967; Wittmann, 1992; Lieder, 2017) and James–Stein estimator (Stein, 1956; James & Stein, 1961); these methods pull/shrink the iterates/estimator towards a specified point $z_0$.

## 4.2  DISCRETE-TIME CONVERGENECE ANALYSIS

We now present convergence results with anchoring. In Theorem 3, we use deterministic gradients, and in Theorem 4, we use stochastic **sub**gradients.

**Theorem 3.**  *Assume* (A0) *and* (A3)*. If* $p \in (1/2, 1)$ *and* $\gamma \geq 2$*, then SimGD-A converges in the sense of*

$$\|G(z_k)\|^2 \leq \mathcal{O}\left(\frac{1}{k^{2-2p}}\right).$$

The proof can be considered a discretization of the continuous-time analysis.

Consider the setup of Section 2. We propose *Anchored Simultaneous Stochastic SubGradient Descent*

$$z_{k+1} = z_k - \frac{1-p}{(k+1)^p}g(z_k; \omega_k) + \frac{(1-p)\gamma}{(k+1)^{1-\varepsilon}}(z_0 - z_k) \qquad \text{(SSSGD-A)}$$

(The small $\varepsilon > 0$ is introduced for the proof of Theorem 4. See Section A for further discussion.)

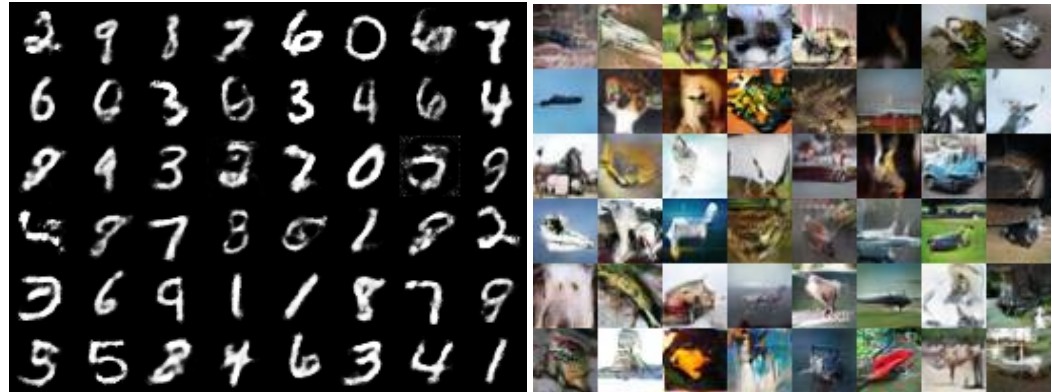

Figure 3: Samples of generated MNIST and CIFAR-10 images at the end of the training periods of anchored Adam.

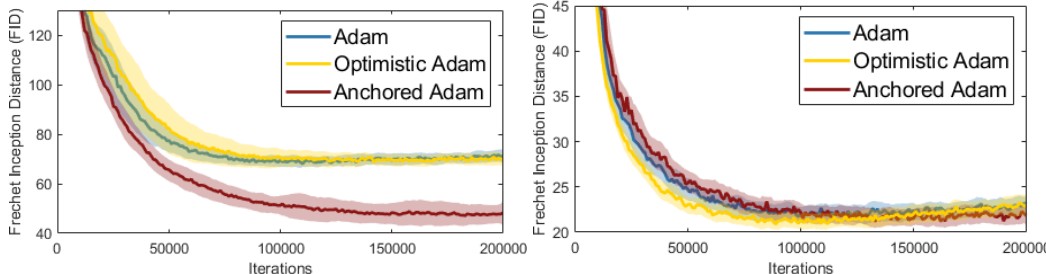

Figure 4: FID score vs. iteration on MNIST (left) and CIFAR-10 (right). Optimism rate of $\beta = 1$ and anchor rate of $\gamma = 1$ was used. The MNIST setup benefits from optimism but not from anchoring, while the CIFAR-10 setup benefits from optimism but not from anchoring.

**Theorem 4.** *Assume* (A0) *and* (A2). *If* $p \in (1/2, 1)$, $\varepsilon \in (0, 1/2)$, *and* $\gamma > 0$, *then SSSGD-A converges in the sense of* $z_k \xrightarrow{L^2} z_\star$, *where* $z_\star$ *is a saddle point.*

(To clarify, we do not assume $L$ is differentiable.)

**Main contribution.** To the best of our knowledge, Theorem 4 is the first result establishing last-iterate convergence for convex-concave cost functions using stochastic subgradients without assuming strict convexity or analogous assumptions.

## 5 EXPERIMENTS

In this section, we experimentally demonstrate the effectiveness of optimism and anchoring for training GANs. We train Wasserstein-GANs (Arjovsky et al., 2017) with gradient penalty (Gulrajani et al., 2017) on the MNIST and CIFAR-10 dataset and plot the Fréchet Inception Distance (FID) (Heusel et al., 2017; Lucic et al., 2018). The experiments were implemented in PyTorch (Paszke et al., 2017). We combine Adam with optimism and anchoring (described precisely in Appendix G) and compare it against the baseline Adam optimizer (Kingma & Ba, 2015). The generator and discriminator architectures and the hyperparameters are described in Appendix G. For optimistic and anchored Adam, we roughly tune the optimism and anchor rates and show the curve corresponding to the best parameter choice.

Figure 4 shows that the MNIST setup benefits from anchoring but not from optimism, while the CIFAR-10 setup benefits from optimism but not from anchoring. We leave comparing the effects of optimism and anchoring in practical GAN training (where the cost function is not convex-concave) as a topic of future work.

## 6 CONCLUSION

In this work, we analyzed the convergence of SSSGD, Optimistic simGD, and Anchored SSSGD. Under the assumption that the cost $L$ is convex-concave, Anchored SSSGD provably converges under the most general setup. Through experiments, we showed that the practical GAN training benefits from optimism and anchoring in some (but not all) setups.

Generalizing these results to accommodate projections and proximal operators, analogous to projected and proximal gradient methods, is an interesting direction of future work. Weight clipping (Arjovsky et al., 2017) and spectral normalization (Miyato et al., 2018) are instances where projections are used in training GANs.

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

## A   FURTHER DISCUSSION ON THE CONVERGENCE RESULTS

Theorems 1, 2, 3, and 4 use related but different notions of convergence. Theorems 1 and 4 are asymptotic (has no rate) while Theorems 2 and 3 are non-asymptotic (has a rate). Theorems 1 and 3 respectively show almost sure and $L^2$ convergence of the iterates. Theorems 2 and 3 show convergence of the squared gradient norm for the best and last iterates, respectively. We did not make these choices. The choices were dictated by what we can prove based on the analysis.

The discrete-time analysis of SimGD-O of Theorem 2 bounds the squared gradient norm of the *best iterate*, while the continuous-time analysis bounds the squared gradient norm of the "last iterate" (at terminal time). The discrepancy comes from the fact that while we have monotonic decrease of $\|g(t)\|$ in continuous-time, we have no analogous monotonicity condition on $\|g_k\|$ in discrete-time. To the best of our knowledge, there is no result establishing a $\mathcal{O}(1/k)$ rate on the squared gradient norm of the *last iterate* for SimGD-O or the related "extragradient method" Korpelevich (1976). Theorem 3 is the first result showing a rate close to $\mathcal{O}(1/k)$ on the last literate.

For SimGD-O and Corollary 1, the parameter choices are almost optimal. The optimal choices that minimize the bound of Theorem 2 are $\alpha = 0.124897/R$ and $\beta = 1.94431\alpha$; they provide a factor of 135.771, a very small improvement over the factor 136 of Corollary 1.

For SimGD-A and Theorem 3, there is a discrepancy in the rate between the continuous time analysis $\mathcal{O}(1/t^2)$ and the discrete time rate $\mathcal{O}(1/k^{2-2p})$ for $p \in (1/2, 1)$, which is slightly slower than $\mathcal{O}(1/k)$. In discretizing the continuous-time calculations to obtain a discrete proof, errors accumulate and prevent the rate from being better than $\mathcal{O}(1/k)$. This is not an artifact of the proof. Simple tests on bilinear examples show divergence when $p < 1/2$.

SSSGD-A and Theorem 4 involves the parameter $\varepsilon$. While the proof requires $\varepsilon > 0$, we believe this is an artifact of the proof. In particular, we conjecture that Lemma 17 holds with $o(s/\tau)$ rather than $\mathcal{O}(s/\tau)$, and, if so, it is possible to establish convergence with $\varepsilon = 0$.

In Figure 2, it seems that that the choice $\varepsilon = 0$ and $p = 2/3$ is optimal for SSSGD-A. While we do not have a theoretical explanation for this, we point out that this is not surprising as $p = 2/3$ is known to be optimal in stochastic convex minimization (Moulines & Bach, 2011; Taylor & Bach, 2019).

Theorems 2, 3, and 4 extend to monotone operators (Ryu & Boyd, 2016; Bauschke & Combettes, 2017) without any modification to their proofs. In infinite dimensional setups (which is of interest in the field of monotone operators) Theorem 4 establishes strong convergence, while many convergence results (including Theorems 2 and 3) establish weak convergence. However, Theorem 1 does not extend to monotone operators, as the use of the LaSalle–Krasnovskii principle is particular to convex-concave saddle functions.

## B   NOTATION AND PRELIMINARIES

Write $\mathbb{R}_+$ to denote the set of nonnegative real numbers and $\langle \cdot, \cdot \rangle$ to denote inner product, i.e., $\langle u, v \rangle = u^T v$ for $u, v \in \mathbb{R}^{m+n}$.

We say $A$ is a point-to-set mapping on $\mathbb{R}^d$ if $A$ maps points of $\mathbb{R}^d$ to subsets of $\mathbb{R}^d$. For notational simplicity, we write

$$\langle A(x) - A(y), x - y \rangle = \{ \langle u - v, x - y \rangle \mid u \in A(x), \, v \in A(y) \}.$$

Using this notation, we define monotonicity of $A$ with

$$\langle A(x) - A(y), x - y \rangle \geq 0 \quad \forall x, y \in \mathbb{R}^d,$$

where the inequality requires every member of the set to be nonnegative. We say a monotone operator $A$ is maximal if there is no other monotone operator $B$ such that the containment

$$\{ (x, u) \mid u \in A(x) \} \subset \{ (x, u) \mid u \in B(x) \}$$

is proper. If $L : \mathbb{R}^m \times \mathbb{R}^n \to \mathbb{R}$ is convex-concave, then the subdifferential operator

$$G(x, u) = \begin{bmatrix} \partial_x L(x, u) \\ \partial_u(-L)(x, u) \end{bmatrix}$$

is maximal monotone (Rockafellar, 1970). By Bauschke & Combettes (2017) Proposition 20.36, $G(z)$ is closed-convex for any $z \in \mathbb{R}^{m+n}$. By Bauschke & Combettes (2017) Proposition 20.38(iii), maximal monotone operators are upper semicontinuous in the sense that if $G$ is maximal monotone, then $g_k \in G(z_k)$ for $k = 0, 1, \dots$ and $(z_k, g_k) \to (z_\infty, g_\infty)$ imply $g_\infty \in G(z_\infty)$. (In other words, the graph of $G$ is closed.) Define $\mathrm{Zer}(G) = \{z \in \mathbb{R}^d \,|\, 0 \in G(z)\}$, which is the set of saddle-points or equilibrium points. When $G$ is maximal monotone, $\mathrm{Zer}(G)$ is a closed convex set. Write

$$P_{\mathrm{Zer}(G)}(z_0) = \arg\min_{z \in \mathrm{Zer}(G)} \|z - z_0\|$$

for the projection onto $\mathrm{Zer}(G)$.

Write $\mathcal{C}(\mathbb{R}_+, \mathbb{R}^d)$ for the space of $\mathbb{R}^d$-valued continuous functions on $\mathbb{R}_+$. For $f_k : \mathbb{R}_+ \to \mathbb{R}^{m+n}$, we say $f_k \to f$ in $\mathcal{C}(\mathbb{R}_+, \mathbb{R}^d)$ if $f_k \to f$ uniformly on bounded intervals, i.e., for all $T < \infty$, we have

$$\lim_{k \to \infty} \sup_{t \in [0,T]} \|f_k(t) - f(t)\| = 0.$$

In other words, we consider the topology of uniform convergence on compact sets.

We rely on the following inequalities, which hold for any $a, b \in \mathbb{R}^{m+n}$ any $\varepsilon > 0$.

$$\langle a, b \rangle \leq \frac{1}{2\varepsilon}\|a\|^2 + \frac{\varepsilon}{2}\|b\|^2 \tag{5}$$

$$\|a + b\|^2 \leq 2\|a\|^2 + 2\|b\|^2. \tag{6}$$

Both inequalities are called Young's inequality. (Note, (6) follows from (5) with $\varepsilon = 1$.)

**Lemma 1** (Theorem 5.3.33 of Dembo (2019)). *Let $\{\mathcal{F}_k\}_{k \in \mathbb{N}_+}$ be an increasing sequence of $\sigma$-algebras. Let $(m_k, \mathcal{F}_k)$ be a martingale such that*

$$\mathbb{E}[\|m_k\|^2] < \infty$$

*for all $k \geq 0$ and*

$$\sum_{k=0}^{\infty} \mathbb{E}\left[\|m_{k+1} - m_k\|^2 \,|\, \mathcal{F}_k\right] < \infty$$

*then $m_k$ converges almost surely to a limit.*

**Lemma 2** (Robbins & Siegmund (1971)). *Let $\{\mathcal{F}_k\}_{k \in \mathbb{N}_+}$ be an increasing sequence of $\sigma$-algebras. Let $\{V_k\}_{k \in \mathbb{N}_+}$, $\{S_k\}_{k \in \mathbb{N}_+}$, $\{U_k\}_{k \in \mathbb{N}_+}$, and $\{\beta_k\}_{k \in \mathbb{N}_+}$ be nonnegative $\mathcal{F}_k$-measurable random sequences satisfying*

$$\mathbb{E}\left[V_{k+1} \,|\, \mathcal{F}_k\right] \leq (1 + \beta_k)V_k - S_k + U_k.$$

*If*

$$\sum_{k=1}^{\infty} \beta_k < \infty, \qquad \sum_{k=1}^{\infty} U_k < \infty$$

*holds almost surely, then*

$$V_k \to V_\infty, \qquad S_k \to 0$$

*almost surely, where $V_\infty$ is a random limit.*

Define

$$\tilde{G}(z) = \mathbb{E}_\omega g(z; \omega) \in G(z).$$

Note that $0 \neq \tilde{G}(z_\star)$ is possible even if $0 \in G(z_\star)$ when $L$ is not continuously differentiable.

**Lemma 3.** *Under Assumptions* (A0) *and* (A2)*, we have*

$$\mathbb{E}_\omega \|g(z; \omega)\|^2 \leq R_3^2 \|z - z_\star\|^2 + R_4^2$$

*for some $R_3 > 0$ and $R_4 > 0$.*

*Proof.* Let $z_\star$ be a saddle point, which exists by Assumption (A0). Let $\omega$ and $\omega'$ be independent and identically distributed. Then

$$
\begin{aligned}
\mathbb{E}_\omega \|g(z;\omega)\|^2 &\leq \mathbb{E}_\omega \|g(z;\omega)\|^2 + \mathbb{E}_{\omega'} \|g(z_\star;\omega') - \tilde{G}(z_\star)\|^2 \\
&= \mathbb{E}_{\omega,\omega'} \|g(z;\omega) - g(z_\star;\omega') + \tilde{G}(z_\star)\|^2 \\
&\leq \mathbb{E}_{\omega,\omega'} 2\|g(z;\omega) - g(z_\star;\omega')\|^2 + 2\|\tilde{G}(z_\star)\|^2 \\
&\leq 2R_1^2 \|z - z_\star\|^2 + 2R_2^2 + 2\|\tilde{G}(z_\star)\|^2
\end{aligned}
$$

where we use the fact that $g(z_\star;\omega') - \tilde{G}(z_\star)$ is a zero-mean random variable, Assumption (A2), and (6). The stated result holds with $R_3^2 = 2R_1^2$ and $R_4^2 = 2R_2^2 + 2\|\tilde{G}(z_\star)\|^2$. $\qquad\square$

## C    ANALYSIS OF THEOREM 1

For convenience, we restate the update, assumptions, and the theorem:

$$
z_{k+1} = z_k - \alpha_k g(z_k;\omega_k) \tag{SSSGD}
$$

$$
L \text{ is convex-concave and has a saddle point} \tag{A0}
$$

$$
\sum_{k=0}^{\infty} \alpha_k = \infty, \qquad \sum_{k=0}^{\infty} \alpha_k^2 < \infty \tag{A1}
$$

$$
\mathbb{E}_{\omega_1,\omega_2} \|g(z_1;\omega_1) - g(z_2;\omega_2)\|^2 \leq R_1^2 \|z_1 - z_2\|^2 + R_2^2 \quad \forall\, z_1, z_2 \in \mathbb{R}^{m+n}, \tag{A2}
$$

**Theorem 1.** *Assume* (A0)*,* (A1)*, and* (A2)*. Furthermore, assume $L(x,u)$ is strictly convex in $x$ for all $u$. Then SSSGD converges in the sense of $z_k \overset{a.s.}{\to} z_\star$ where $z_\star$ is a saddle point of $L$.*

**Differential inclusion technique.**    We use the differential inclusion technique of Duchi & Ruan (2018), also recently used in Davis et al. (2019). The high-level summary of the technique is very simple and elegant: (i) show the discrete-time process converges to a continuous-time trajectory satisfying a differential inclusion, (ii) show any solution of the differential inclusion has a desirable property, and (iii) translate the conclusion in continuous-time to discrete-time. However, the actual execution of this technique does require careful and technical considerations.

**Proof outline.**    For step (i), we adapt the LaSalle–Krasnovskii principle to show that a solution of the continuous-time differential inclusion converges to a saddle point. (Lemma 5.) Then we carry out step (ii) showing the time-shifted interpolated discrete time process converges to a solution of the differential inclusion. (Lemma 6.) Finally, step (iii), the "Continuous convergence to discrete convergence", combines these two pieces to conclude that the discrete time process converges to a saddle point. The contribution and novelty of our proof is in our steps (i) and (iii).

**Preliminary definitions and results.**    Consider the *differential inclusion*

$$
\dot{z}(t) \in -G(z(t)) \tag{7}
$$

with the initial condition $z(0) = z_0$. We say $z : [0,\infty) \to \mathbb{R}^{m+n}$ satisfies (7) if there is a Lebesgue integrable $\zeta : [0,\infty) \to \mathbb{R}^{m+n}$ such that

$$
z(t) = z_0 + \int_0^t \zeta(s)\, ds, \qquad \zeta(t) \in -G(z(t)), \forall\, t \geq 0. \tag{8}
$$

Write $z(t) = \phi_t(z_0)$ and call $\phi_t : \mathbb{R}^{m+n} \to \mathbb{R}^{m+n}$ the *time evolution operator*. In other words, $\phi_t$ maps the initial condition of the differential inclusion to the point at time $t$, which is well defined by the following result.

**Lemma 4** (Theorem 5.2.1 of Aubin & Cellina (1984))**.** *If $G$ is maximal monotone, the solution to* (7) *exists and is unique. Furthermore, $\phi_t : \mathbb{R}^{m+n} \to \mathbb{R}^{m+n}$ is 1-Lipschitz continuous for all $t \geq 0$.*

## C.1 PROOF OF THEOREM 1

Lemma 5 and its proof can be considered an adaptation of the LaSalle–Krasnovskii invariance principle (Krasovskii, 1959; LaSalle, 1960) to the setup of differential inclusions. The standard result applies to differential equations.

**Lemma 5** (LaSalle–Krasnovskii). *Assume* (A0). *Assume $L(x, u)$ is strictly convex in $x$ for all $u$. If $z(\cdot)$ satisfies* (7)*, then $z(t) \to z_\infty$ as $t \to \infty$ and $z_\infty \in \text{Zer}(G)$.*

*Proof.* Consider any $z_\star \in \text{Zer}(G)$, which exists by Assumption (A0). Since $z(t)$ is absolutely continuous, so is $\|z(t) - z_\star\|^2$, and we have

$$\frac{d}{dt} \frac{1}{2} \|z(t) - z_\star\|^2 = \langle \zeta(t), z(t) - z_\star \rangle \leq 0$$

for almost all $t > 0$, where $\zeta(\cdot)$ is as defined in (8) and the inequality follows from (1), monotonicity of $G$. Therefore, $\|z(t) - z_\star\|^2$ is a nonincreasing function of $t$, and

$$\lim_{t \to \infty} \|z(t) - z_\star\| = \chi$$

for some limit $\chi \geq 0$. Since $z(t)$ is a bounded sequence, it has at least one cluster point.

Let $t_k \to \infty$ such that $z(t_k) \to z_\infty$, i.e., $z_\infty$ is a cluster point of $z(\cdot)$. Then, $\|z_\infty - z_\star\|^2 = \chi$. Since $\phi_t(\cdot)$ (with fixed $t$) is continuous by Lemma 4, we have

$$\lim_{k \to \infty} \phi_{s+t_k}(z_0) = \lim_{k \to \infty} \phi_s(\phi_{t_k}(z(0))) = \phi_s(z_\infty)$$

for all $s \geq 0$. This means $\phi_s(z_\infty)$ is also a cluster point of $z(\cdot)$ and

$$\|\phi_s(z_\infty) - z_\star\| = \chi$$

for all $s \geq 0$. Therefore

$$0 = \frac{d}{ds} \|\phi_s(z_\infty) - z_\star\|^2 \in -\langle G(\phi_s(z_\infty)), \phi_s(z_\infty) - z_\star \rangle \tag{9}$$

for almost all $s \geq 0$.

Write $z_\infty = (x_\infty, u_\infty)$ and let $z_\star = (x_\star, u_\star) \in \text{Zer}(G)$. Write $(\phi_s^x(z_\star), \phi_s^u(z_\star)) = (\phi_s(z_\star))$. If $\phi_s^x(z_\star) \neq x_\star$

$$\langle G(\phi_s(z_\infty)), \phi_s(z_\infty) - z_\star \rangle > 0$$

by strict convexity, and, in light of (9), we conclude $\phi_s^x(z_\star) = x_\star$ for almost all $s \geq 0$. Then for almost all $s \geq 0$, we have

$$\begin{aligned}
0 &\in \langle G(\phi_s(z_\infty)), \phi_s(z_\infty) - z_\star \rangle \\
&= \langle \partial_u(-L)(x_\star, \phi_s^u(z_\infty)), \phi_s^u(z_\infty) - u_\star \rangle \\
&\geq -L(x_\star, \phi_s^u(z_\infty)) + L(x_\star, u_\star) \\
&\geq 0,
\end{aligned}$$

where the first inequality follows from concavity of $L(x, u)$ in $u$ and the second inequality follows from the fact that $u_\star$ is a maximizer when $x_\star$ is fixed. Therefore, we have equality throughout, and $L(x_\star, \phi_s^u(z_\infty)) = L(x_\star, u_\star)$, i.e., $\phi_s^u(z_\infty)$ also maximizes $L(x_\star, \cdot)$.

Remember that $\phi_s(z_\infty)$ is a continuous function of $s$ for all $s \geq 0$. Therefore, that $\phi_s^x(z_\infty) = x_\star$ and that $\phi_s^u(z_\infty)$ maximizes $L(x_\star, \cdot)$ for almost all $s \geq 0$ imply that the conditions hold for $s = 0$. In other words, $x_\infty = x^\star$ and $u_\infty$ maximizes $L(x_\star, \cdot)$, and therefore $z_\infty \in \text{Zer} G$.

Finally, since $z_\infty$ is a solution, $\|z(t) - z_\infty\|$ converges to a limit as $t \to \infty$. Since $\|z(t_k) - z_\infty\| \to 0$, we conclude that $\|z(t) - z_\infty\| \to 0$ as $t \to \infty$. □

The following lemma is the crux of the differential inclusion technique. It makes precise in what sense the discrete-time process converges to a solution of the continuous-time differential inclusion.

**Lemma 6** (Theorem 3.7 of Duchi & Ruan (2018)). *Consider the update*

$$z_{k+1} = z_k - \alpha_k(\zeta_k + \xi_k), \qquad \zeta_k \in G(z_k).$$

*Define $t_k = \sum_{i=1}^{k} \alpha_i$ and*

$$z_{\text{interp}}(t) = z_k + \frac{t - t_k}{t_{k+1} - t_k}(z_{k+1} - z_k), \quad t \in [t_k, t_{k+1}).$$

*Define the time-shifted process*

$$z_{\text{interp}}^{\tau}(\cdot) = z_{\text{interp}}(\tau + \cdot).$$

*Let the following conditions hold:*

(i) *The iterates are bounded, i.e., $\sup_k \|z_k\| < \infty$ and $\sup_k \|\zeta_k\| < \infty$.*

(ii) *The stepsizes $\alpha_k$ satisfy Assumption (A1).*

(iii) *The weighted noise sequence converges: $\sum_{k=0}^{\infty} \alpha_k \xi_k = v$ for some $v \in \mathbb{R}^d$.*

(iv) *For any increasing sequence $n_k$ such that $z_{n_k} \to z_\infty$, we have*

$$\lim_{n \to \infty} \text{dist}\left(\frac{1}{m}\sum_{k=1}^{m} \zeta_{n_k}, G(z_\infty)\right) = 0.$$

*Then for any sequence $\{\tau_k\}_{k=1}^{\infty} \subset \mathbb{R}_+$, the sequence of functions $\{z_{\text{interp}}^{\tau_k}(\cdot)\}$ is relatively compact in $\mathcal{C}(\mathbb{R}_+, \mathbb{R}^d)$. If $\tau_k \to \infty$, all cluster points of $\{z_{\text{interp}}^{\tau_k}(\cdot)\}$ satisfy the differential inclusion (8).*

We verify the conditions of Lemma 6 and make the argument that the noisy discrete time process is close to the noiseless continuous time process and the two processes converge to the same limit.

**Verifying conditions of Lemma 6.**
*Condition (i).* Let $z_\star \in \text{Zer}(G)$. Write $\mathcal{F}_k$ for the $\sigma$-field generated by $\omega_0, \ldots, \omega_{k-1}$. Write $\tilde{G}(z) = \mathbb{E}g(z; \omega) \in G(z)$. Then

$$\|z_{k+1} - z_\star\|^2 = \|z_k - z_\star\|^2 - 2\alpha_k\langle z_k - z_\star, g(z_k; \omega_k)\rangle + \alpha_k^2\|g(z_k; \omega_k)\|^2$$

$$\mathbb{E}\left[\|z_{k+1} - z_\star\|^2 \mid \mathcal{F}_k\right] \leq \|z_k - z_\star\|^2 - 2\alpha_k\langle z_k - z_\star, \tilde{G}(z_k)\rangle + \alpha_k^2\left(R_3^2\|z_k - z_\star\|^2 + R_4^2\right)$$

$$= (1 + \alpha_k^2 R_3^2)\|z_k - z_\star\|^2 - 2\alpha_k\langle z_k - z_\star, \tilde{G}(z_k)\rangle + \alpha_k^2 R_4^2,$$

where we used Assumption (A2) and Lemma 3. Since $\sum_{k=0}^{\infty} \alpha_k^2 < \infty$ by Assumption (A1), this inequality and Lemma 2 tells us

$$\|z_k - z_\star\|^2 \to \text{limit}$$

for some limit, which implies $z_k$ is a bounded sequence. Since $z_k$ is bounded, so is $\tilde{G}(z_k)$ since

$$\|\tilde{G}(z_k)\|^2 \leq \mathbb{E}_\omega\|g(z_k; \omega)\|^2 \leq R_3^2 \sup_k \|z_k - z_\star\|^2 + R_4^2$$

by Lemma 3.

*Condition (ii).* This condition is assumed.

*Condition (iii).* Define

$$\xi^k = g(z_k; \omega_k) - \tilde{G}(z_k)$$

and

$$m_k = \sum_{i=0}^{k} \alpha_i \xi_i.$$

Then $(m_k, \mathcal{F}_k)$ is a martingale and

$$
\begin{aligned}
\sum_{k=0}^{\infty} \mathbb{E}\left[\|m_{k+1} - m_k\|^2 \,|\, \mathcal{F}_k\right] &= \sum_{k=0}^{\infty} \alpha_k^2 \mathbb{E}\left[\|\xi_k\|^2 \,|\, \mathcal{F}_k\right] \\
&\leq \sum_{k=0}^{\infty} \alpha_k^2 \mathbb{E}\left[\|g(z_k; \omega_k)\|^2 \,|\, \mathcal{F}_k\right] \\
&\leq \sum_{k=0}^{\infty} \alpha_k^2 \left(R_3^2 \|z_k - z_\star\|^2 + R_4^2\right) \\
&\leq \sum_{k=0}^{\infty} \alpha_k^2 \left(\sup_k 2R_3^2\|z_k\| + 2R_3^2\|z_\star\|^2 + R_4^2\right) < \infty
\end{aligned}
$$

almost surely, where the first inequality is the second moment upper bounding the variance, the second inequality is Lemma 3, and the third inequality is (6) and condition (i). Finally, we have (iii) by Lemma 1.

*Condition (iv).* As discussed in Section B, $G$ is maximal monotone, which implies $G$ is upper semicontinuous, i.e., $(z_{n_k}, g_{n_k}) \to (z_\infty, g_\infty)$ implies $g_\infty \in G(z_\infty)$, and $G(z_\infty)$ is a closed convex set. Therefore, $\mathrm{dist}(\zeta_{n_k}, G(z_\infty)) \to 0$ as otherwise we can find a further subsequence such that converging to $\zeta_\infty$ such that $\mathrm{dist}(\zeta_\infty, G(z_\infty)) > 0$. (Here we use the fact that $\zeta_k$ is bounded due to condition (i)). Since $G(z_\infty)$ is a convex set,

$$
\mathrm{dist}(\zeta_{n_k}, G(z_\infty)) \to 0 \Rightarrow \frac{1}{m}\sum_{k=1}^{m} \mathrm{dist}(\zeta_{n_k}, G(z_\infty)) \to 0 \Rightarrow \mathrm{dist}\left(\frac{1}{m}\sum_{k=1}^{m} \zeta_{n_k}, G(z_\infty)\right) \to 0.
$$

In the main proof, we show that cluster points of $z_{\mathrm{interp}}(\cdot)$ are solutions. We need the following lemma to conclude that these cluster points are also cluster points of the original discrete time process $z_k$.

**Lemma 7.** *Under the conditions of Lemma 6, $z_{\mathrm{interp}}(\cdot)$ and $z_k$ share the same cluster points.*

*Proof.* If $z_\infty$ is a cluster point of $z_k$, then it is a cluster point of $z_{\mathrm{interp}}(\cdot)$ by definition. Assume $z_\infty$ is a cluster point of $z_{\mathrm{interp}}(\cdot)$, i.e., assume there is a sequence $\tau_j \to \infty$ such that $z_{\mathrm{interp}}(\tau_j) \to z_\infty$. Define $k_j \to \infty$ with

$$
t_{k_j} \leq \tau_j < t_{k_j+1}.
$$

Then

$$
\begin{aligned}
\|z_{\mathrm{interp}}(\tau_j) - z_{k_j}\| &\leq \alpha_k \|z_{k_j+1} - z_{k_j}\| \\
&\leq \alpha_k (\|\zeta_k\| + \|\xi_k\|) \\
&\to 0
\end{aligned}
$$

where we use the assumption (i) which states that $\|\zeta_k\|$ is bounded and assumption (iii) which states that $\alpha_k \xi_k \to 0$. We conclude $z_{k_j} \to z_\infty$. $\qquad\square$

**Continuous convergence to discrete convergence.** Let $k_j \to \infty$ be a subsequence such that $z_{k_j} \to z_\infty$. Let $k_j' \to \infty$ be a further subsequence such that

$$
\lim_{k_j' \to \infty} z_{\mathrm{interp}}^{t_{k_j'}}(T) = \phi_T(z_\infty)
$$

for all $T \geq 0$, which exists by Lemma 6. (The time-shifted interpolated process converges to a solution of the differential inclusion.) By Lemma 5,

$$
\lim_{T \to \infty} \phi_T z_\infty \to \phi_\infty z_\infty
$$

where $\phi_t z_\infty \to \phi_\infty z_\infty$ as $t \to \infty$ and $\phi_\infty z_\infty$ is a saddle point. (The solution to the differential inclusion converges to a solution.)

These facts together imply that for any $\varepsilon > 0$, there exists $k_j'$ and $\tau_j$ large enough that

$$\|z_{\text{interp}}^{t_{k_j'}}(\tau_j) - \phi_{\tau_j}(z_\infty)\| < \varepsilon/2$$

and

$$\|\phi_{\tau_j} z_\infty - \phi_\infty z_\infty\| < \varepsilon/2.$$

Together, these imply

$$\|z_{\text{interp}}(t_{k_j'} + \tau_j) - \phi_\infty z_\infty\| < \varepsilon.$$

since $z_{\text{interp}}^\tau(\cdot) = z_{\text{interp}}(\tau + \cdot)$. Therefore, $\phi_\infty z_\infty$ is a cluster point of $z_{\text{interp}}(\cdot)$, and, by Lemma 7, $\phi_\infty z_\infty$ is a cluster point of $z_k$.

Since $\|z_k - \phi_\infty z_\infty\|$ converges to a limit and converges to 0 on this further subsequence, we conclude $\|z_k - \phi_\infty z_\infty\| \to 0$ almost surely. $\square$

## D   ANALYSIS OF THEOREM 2

For convenience, we restate the update, assumptions, and the theorem:

$$z_{k+1} = z_k - \alpha G(z_k) - \beta(G(z_k) - G(z_{k-1})) \tag{SimGD-O}$$

$$L \text{ is convex-concave and has a saddle point} \tag{A0}$$

$$L \text{ is differentiable and } \nabla L \text{ is } R\text{-Lipschitz continuous} \tag{A3}$$

**Theorem 2.**   *Assume* (A0) *and* (A3). *If* $0 < \alpha < 2\beta(1 - 2\beta R)$, *then SimGD-O converges in the sense of*

$$\min_{i=0,\dots,k} \|G(z_k)\|^2 \le \frac{2 + 2\beta^2 R^2}{\alpha(2\beta - \alpha - 4\beta^2 R)k} \|z_0 + \beta G(z_0) - z_\star\|^2.$$

*Furthermore,* $z_k \to z_\star$, *where* $z_\star$ *is a saddle point of L.*

### D.1   PROOF OF THEOREM 2

Throughout this section, write $g_k = G(z_k)$ for $k \ge -1$. Since we can define $\tilde{G} = \alpha G$ and $\tilde{\beta} = \beta/\alpha$ and write the iteration as

$$z_{k+1} = z_k - \tilde{G}(z_k) - \tilde{\beta}(\tilde{G}(z_k) - \tilde{G}(z_{k-1})),$$

we assume $\alpha = 1$ without loss of generality. Then

$$\|z_{k+1} + \beta g_k - z_\star\|^2 = \|z_k + \beta g_{k-1} - z_\star\|^2 - 2\langle g_k, z_k - z_\star \rangle - \langle g_k, 2\beta g_{k-1} - g_k \rangle$$
$$\le \|z_k + \beta g_{k-1} - z_\star\|^2 - \langle g_k, 2\beta g_{k-1} - g_k \rangle,$$

where the inequality follows from (1), monotonicity of $G$, and

$$-\langle g_k, 2\beta g_{k-1} - g_k \rangle = 4\beta^2 \langle g_k - g_{k-1}, z_k - z_{k+1} \rangle - (2\beta - 1)\|z_{k+1} - z_k\|^2 - \beta^2(1 + 2\beta)\|g_k - g_{k-1}\|^2$$
$$\le 4\beta^2 \langle g_k - g_{k-1}, z_k - z_{k+1} \rangle - (2\beta - 1)\|z_{k+1} - z_k\|^2.$$

We can bound

$$4\beta^2 \langle g_k - g_{k-1}, z_k - z_{k+1} \rangle \le \frac{2\beta^2}{R}\|g_k - g_{k-1}\|^2 + 2\beta^2 R\|z_{k+1} - z_k\|^2$$
$$\le 2\beta^2 R\|z_k - z_{k-1}\|^2 + 2\beta^2 R\|z_{k+1} - z_k\|^2,$$

where the first inequality follows from (5), Young's inequality, with $\varepsilon = R$ and the second inequality follows from Assumption (A3), $R$-Lipschitz continuity of $G$. Putting these together we get

$$\|z_{k+1} + \beta g_k - z_\star\|^2 \le \|z_k + \beta g_{k-1} - z_\star\|^2$$
$$+ 2\beta^2 R\|z_k - z_{k-1}\|^2 - (2\beta - 1 - 2\beta^2 R)\|z_{k+1} - z_k\|^2. \tag{10}$$

Since $\beta > 1/2$ and $R < (2\beta - 1)/(4\beta^2)$ is assumed for Theorem 2, we have

$$2\beta^2 R < (2\beta - 1 - 2\beta^2 R).$$

By summing (10), we have

$$\left(2\beta - 1 - 2\beta^2 R\right)\sum_{i=0}^{k}\|z_{i+1} - z_i\|^2 - 2\beta^2 R\sum_{i=0}^{k}\|z_i - z_{i-1}\|^2 \leq \|z_0 + \beta g_{-1} - z_\star\|^2$$

$$\left(2\beta - 1 - 4\beta^2 L\right)\sum_{i=0}^{k}\|z_{i+1} - z_i\|^2 \leq \|z_0 + \beta g_{-1} - z_\star\|^2, \tag{11}$$

where we use $z_0 = z_{-1}$.

Next,

$$\|g_k\|^2 = \|z_{k+1} - z_k + \beta(g_k - g_{k-1})\|^2$$
$$\leq 2\|z_{k+1} - z_k\|^2 + 2\beta^2\|g_k - g_{k-1}\|^2$$
$$\leq 2\|z_{k+1} - z_k\|^2 + 2\beta^2 R^2\|z_k - z_{k-1}\|^2,$$

where we use (6). Using (11), we get

$$\sum_{i=1}^{k}\left(2\|z_{i+1} - z_i\|^2 + 2\beta^2 R^2\|z_i - z_{i-1}\|^2\right) \leq \frac{2 + 2\beta^2 R^2}{2\beta - 1 - 4\beta^2 R}\|z_0 + \beta g_{-1} - z_\star\|^2.$$

Therefore, $2\|z_{k+1} - z_k\|^2 + 2\beta^2 R^2\|z_k - z_{k-1}\|^2 \to 0$ and $\|g_k\|^2 \to 0$. Moreover, we have

$$\min_{i=0,\ldots,k}\|g_i\|^2 \leq \frac{2 + 2\beta^2 R^2}{(2\beta - 1 - 4\beta^2 R)k}\|z_0 + \beta g_{-1} - z_\star\|^2.$$

By scaling $G$ by $\alpha$, we get the first stated result.

By summing (10), we have

$$\|z_k + \beta g_{k-1} - z_\star\|^2 \leq \|z_0 + \beta g_{-1} - z_\star\|^2,$$

and using the triangle inequality we get

$$\|z_k - z_\star\| \leq \|z_0 + \beta g_{-1} - z_\star\| + \beta\|g_{k-1}\| \to \|z_0 + \beta g_{-1} - z_\star\|$$

as $k \to \infty$. (Remember $g_k \to 0$.) So $z_k$ is a bounded sequence, and let $z_\infty$ be the limit of a convergent subsequence $z_{n_k}$. Since $G$ is a continuous mapping with $g_{n_k} = G(z_{n_k})$, $z_{n_k} \to z_\infty$, and $g_{n_k} \to 0$, we have $G(z_\infty) = 0$.

Finally, we show that the entire sequence $z_k$ converges to $z_\infty$. Reorganizing (10), we get

$$\|z_{k+1} + \beta g_k - z_\star\|^2 + 2\beta^2 R\|z_{k+1} - z_k\|^2 \leq \|z_k + \beta g_{k-1} - z_\star\|^2 + 2\beta^2 R\|z_k - z_{k-1}\|^2$$
$$- \underbrace{\left(2\beta - 1 - 4\beta^2 R\right)}_{>0}\|z_{k+1} - z_k\|^2.$$

So $\|z_{k+1} + \beta g_k - z_\star\|^2 + 2\beta^2 R\|z_{k+1} - z_k\|^2$ is a nonincreasing sequence, and the following limit exists

$$\lim_{k\to\infty}\|z_k + \beta g_{k-1} - z_\star\|^2 + 2\beta^2 R\|z_k - z_{k-1}\|^2 = \|z_\infty - z_\star\|^2.$$

Since $z_\star$ can be any equilibrium point, we let $z_\star = z_\infty$. This proves $\|z_k - z_\infty\|^2 \to 0$, i.e., $z_k \to z_\infty$. □

## E  ANALYSIS OF THEOREM 3

### E.1  PRELIMINARY LEMMAS

We quickly state a few identities and inequalities we later use. As the verification of these results are elementary, we only provide a short summary of their proofs.

**Lemma 8.** *For $p \in (0, 1)$ and $k \geq 1$,*

$$\frac{p}{k} - \frac{p(1-p)}{2k^2} < \frac{(k+1)^p - k^p}{k^p} < \frac{p}{k}.$$

The proof follows from a basic application of the inequality

$$1 + px - \frac{p(1-p)}{2}x^2 \le (1+x)^p \le 1 + px$$

for $x \in [0,1]$ and $p \in (0,1)$.

**Lemma 9.** *For $p \in (0,1)$ and $k \ge 1$,*

$$\frac{p}{k+1} < \frac{(k+1)^p - k^p}{k^p}.$$

The proof follows from integrating the decreasing function $p/x^{1-p}$ from $k$ to $k+1$.

**Lemma 10.** *For $p \in (0,1)$ and $k \ge 1$,*

$$0 \le \frac{p}{k(k+1)} - \frac{(k+1)^p - k^p}{k^p(k+1)} \le \frac{p(1-p)}{2k^3}.$$

The proof follows from Lemma 8.

**Lemma 11.** *Given any $V_0, V_1, \ldots \in \mathbb{R}$, we have*

$$\sum_{j=1}^{k} \left( \frac{j(j+1)}{2} (V_j - V_{j-1}) + jV_{j-1} \right) = \frac{k(k+1)}{2} V_k$$

The proof follows from basic calculations. This result can be thought of as the discrete analog of

$$\int_0^t \frac{s^2}{2} \dot{V}(s) + sV(s) \, ds = \frac{t^2 V}{2}.$$

**Lemma 12.** *Let $z_0, z_1, \ldots \in \mathbb{R}^{m+n}$ be an arbitrary sequence. Then for any $k = 0, 1, \ldots,$*

$$\frac{1}{2}\|z_{k+1} - z_0\|^2 - \frac{1}{2}\|z_k - z_0\|^2 = \left\langle z_{k+1} - z_k, \frac{1}{2}(z_{k+1} + z_k) - z_0 \right\rangle.$$

The proof follows from basic calculations. This result can be thought of as the discrete analog of

$$\frac{d}{dt} \frac{1}{2}\|z(t) - z_0\|^2 = \langle \dot{z}(t), z(t) - z_0 \rangle.$$

### E.2 Convergent sequence lemmas

In the proofs of Theorems 3 and 4, we establish certain descent inequalities. The following lemmas state that these inequalities imply boundedness or convergence.

**Lemma 13.** *Let $\{V_k\}_{k \in \mathbb{N}_+}$ and $\{U_k\}_{k \in \mathbb{N}_+}$ be nonnegative (deterministic) sequences satisfying*

$$V_{k+1} \le \left( 1 - \frac{C_1}{k^{1-\varepsilon}} + f(k) \right) V_k + \frac{C_2}{k^{1-\varepsilon}} \sqrt{V_k} + U_k$$

*where $C_1 > 0$, $C_2 > 0$, $f(k) = o(1/k^{1-\varepsilon})$ with $\varepsilon \in [0,1)$, and*

$$\sum_{k=1}^{\infty} U_k < \infty.$$

*Then $\limsup_{k \to \infty} V_k \le C_2^2/C_1^2$.*

*Proof.* For any $\delta \in (0, C_1)$, there is a large enough $K \ge 0$ such that for all $k \ge K$,

$$\frac{C_1}{k^{1-\varepsilon}} - f(k) \ge \frac{C_1 - \delta/2}{k^{1-\varepsilon}}.$$

Define

$$\nu = \frac{C_2^2}{(C_1 - \delta)^2}$$

for $k \geq 0$. Then

$$V_{k+1} \leq \left(1 - \frac{C_1 - \delta/2}{k^{1-\varepsilon}}\right) V_k + \frac{C_2^2}{(C_1 - \delta)k^{1-\varepsilon}} \max\left\{\sqrt{\frac{V_k}{\nu}}, \frac{V_k}{\nu}\right\} + U_k$$

$$V_{k+1} - \nu \leq \left(1 - \frac{C_1 - \delta/2}{k^{1-\varepsilon}}\right) (V_k - \nu) - \frac{C_2^2 \delta}{2k^{1-\varepsilon}(C_1 - \delta)^2}$$

$$+ \frac{C_2^2}{(C_1 - \delta)k^{1-\varepsilon}} \max\left\{\sqrt{\frac{V_k}{\nu}} - 1, \frac{V_k}{\nu} - 1\right\} + U_k$$

Note that $\max\{\sqrt{x} - 1, x - 1\} \leq \max\{0, x - 1\}$ for all $x \geq 0$. So

$$V_{k+1} - \nu \leq \left(1 - \frac{C_1 - \delta/2}{k^{1-\varepsilon}}\right) (V_k - \nu) + \frac{C_2^2}{(C_1 - \delta)k^{1-\varepsilon}} \max\left\{0, \frac{V_k}{\nu} - 1\right\} + U_k$$

$$= \left(1 - \frac{C_1 - \delta/2}{k^{1-\varepsilon}}\right) (V_k - \nu) + \frac{C_1 - \delta}{k^{1-\varepsilon}} \max\{0, V_k - \nu\} + U_k$$

$$\leq \left(1 - \frac{C_1 - \delta/2}{k^{1-\varepsilon}}\right) \max\{0, V_k - \nu\} + \frac{C_1 - \delta}{k^{1-\varepsilon}} \max\{0, V_k - \nu\} + U_k$$

$$= \left(1 - \frac{\delta}{2k^{1-\varepsilon}}\right) \max\{0, V_k - \nu\} + U_k$$

for large enough $k$. Since

$$0 \leq \left(1 - \frac{\delta}{2k^{1-\varepsilon}}\right) \max\{0, V_k - \nu\} + U_k$$

for large enough $k$, we have

$$\max\{0, V_{k+1} - \nu\} \leq \left(1 - \frac{\delta}{2k^{1-\varepsilon}}\right) \max\{0, V_k - \nu\} + U_k$$

With a standard recursion argument (e.g. Lemma 3 of (Polyak, 1987)) we conclude $\max\{0, V_k - \nu\} \to 0$. Since this holds for any $\delta > 0$, we conclude $\limsup_{k \to \infty} V_k \leq C_2^2/C_1^2$. $\quad\square$

**Lemma 14.** *Let $\varepsilon \in (0, 1)$. Let $\{V_k\}_{k \in \mathbb{N}_+}$ and $\{U_k\}_{k \in \mathbb{N}_+}$ be nonnegative (deterministic) sequences satisfying*

$$V_{k+1} \leq \left(1 - \frac{C}{k^{1-\varepsilon}} + f(k)\right) V_k + g(k)\sqrt{V_k} + U_k$$

*where $C > 0$, $f(k) = o(1/k^{1-\varepsilon})$, $g(k) = \mathcal{O}(1/k)$, and*

$$\sum_{k=1}^{\infty} U_k < \infty.$$

*Then $V_k \to 0$.*

*Proof.* For any $\delta > 0$, there is a large enough $K \geq 0$ such that

$$V_{k+1} \leq \left(1 - \frac{C - \delta}{k^{1-\varepsilon}}\right) V_k + \frac{\delta}{k^{1-\varepsilon}} \sqrt{V_k} + U_k$$

for all $k \geq K$. By Lemma 13, we conclude $\limsup_{k \to \infty} V_k \leq \delta^2/(C - \delta)^2$. Since this holds for all $\delta > 0$, we conclude $V_k \to 0$. $\quad\square$

### E.3 PROOF OF THEOREM 3

For convenience, we restate the update, assumptions, and the theorem:

$$z_{k+1} = z_k - \frac{1-p}{(k+1)^p} G(z_k) + \frac{(1-p)\gamma}{k+1}(z_0 - z_k) \tag{SimGD-A}$$

$$L \text{ is convex-concave and has a saddle point} \tag{A0}$$

$$L \text{ is differentiable and } \nabla L \text{ is } R\text{-Lipschitz continuous} \tag{A3}$$

**Theorem 3.** *Assume* (A0) *and* (A3). *If* $p \in (1/2, 1)$ *and* $\gamma \geq 2$, *then SimGD-A converges in the sense of*

$$\|G(z_k)\|^2 \leq \mathcal{O}\left(\frac{1}{k^{2-2p}}\right).$$

**Proof outline.** Lemma 15 shows the iterates $z_k$ are bounded. Lemma 16 shows that $\|z_{k+1} - 2z_k + z_{k-1}\|^2$, the analog of $\|\ddot{z}\|^2$, is small. The second-order derivative $\ddot{z}$ does not arise in the continuous-time analysis of Section 4.1. In the discrete-time setup, $\|z_{k+1} - 2z_k + z_{k-1}\|^2$ does arise, but we use Lemma 16 to show that its contribution is small. The main proof follows by mimicking the continuous-time analysis by bounding the higher-order terms.

Throughout this section, write $g_k = G(z_k)$ for $k \geq -1$.

**Lemma 15.** *For SimGD-A,*

$$\|z_k - z_\star\|^2 \leq C$$

*for all* $k \geq 0$ *for some* $C > 0$. *(This result depends on assumption* $p > 1/2$.*)*

*Proof.*

$$
\begin{aligned}
\|z_{k+1} - z_\star\|^2 &= \|z_k - z_\star\|^2 - \frac{2(1-p)}{(k+1)^p}\langle g_k, z_k - z_\star\rangle + \frac{2\gamma(1-p)}{k+1}\langle z_0 - z_k, z_k - z_\star\rangle \\
&\quad + \left\|\frac{1-p}{(k+1)^p}g_k + \frac{\gamma(1-p)}{k+1}(z_0 - z_k)\right\|^2 \\
&\leq \left(1 - \frac{2\gamma(1-p)}{k+1}\right)\|z_k - z_\star\|^2 + \frac{2\gamma(1-p)}{k+1}\langle z_0 - z_\star, z_k - z_\star\rangle \\
&\quad + \frac{2(1-p)^2}{(k+1)^{2p}}\|g_k\|^2 + \frac{2\gamma^2(1-p)^2}{(k+1)^2}\|z_0 - z_k\|^2 \\
&\leq \left(1 - \frac{2\gamma(1-p)}{k+1} + \frac{4\gamma^2(1-p)^2}{(k+1)^2}\right)\|z_k - z_\star\|^2 + \frac{2\gamma(1-p)}{k+1}\|z_0 - z_\star\|\|z_k - z_\star\| \\
&\quad + \frac{2(1-p)^2}{(k+1)^{2p}}R^2\|z_k - z_0\|^2 + \frac{4\gamma^2(1-p)^2}{(k+1)^2}\|z_0 - z_\star\|^2 \\
&= \left(1 - \frac{2\gamma(1-p)}{k+1} + \frac{4\gamma^2(1-p)^2}{(k+1)^2} + R_1^2\frac{4(1-p)^2}{(k+1)^{2p}}\right)\|z_k - z_\star\|^2 \\
&\quad + \frac{2\gamma(1-p)}{k+1}\|z_0 - z_\star\|\|z_k - z_\star\| + \frac{4\gamma^2(1-p)^2}{(k+1)^2}\|z_0 - z_\star\|^2
\end{aligned}
$$

where the first inequality follows from (1), the monotonicity inequality, and (6) and the second inequality follows from Assumption A3. We conclude the statement with Lemma 13. $\qquad\square$

**Lemma 16.** *For SimGD-A,*

$$
\begin{aligned}
\|z_{k+1} - 2z_k &+ z_{k-1}\|^2 \\
&\leq 4(1-p)^2\left(\frac{\gamma^2}{k^2} + \frac{R^2}{k^{2p}}\right)\|z_k - z_{k-1}\|^2 + 4(1-p)^2\left(\frac{p^2R^2}{k^{2+2p}} + \frac{\gamma^2}{k^4}\right)\|z_0 - z_k\|^2
\end{aligned}
$$

*Proof.*

$$\|z_{k+1} - 2z_k + z_{k-1}\|^2$$

$$= \left\| \frac{1-p}{(k+1)^p} g_k - \frac{1-p}{k^p} g_{k-1} - \frac{(1-p)\gamma}{k+1}(z_0 - z_k) + \frac{(1-p)\gamma}{k}(z_0 - z_{k-1}) \right\|^2$$

$$\leq 2 \left\| \frac{1-p}{(k+1)^p} g_k - \frac{1-p}{k^p} g_{k-1} \right\|^2 + 2 \left\| \frac{\gamma(1-p)}{k+1}(z_0 - z_k) - \frac{\gamma(1-p)}{k}(z_0 - z_{k-1}) \right\|^2$$

$$\leq \frac{4(1-p)^2}{k^{2p}} \|g_k - g_{k-1}\|^2 + 4 \left( \frac{1-p}{(k+1)^p} - \frac{1-p}{k^p} \right)^2 \|g_k\|^2$$

$$+ \frac{4\gamma^2(1-p)^2}{k^2} \|z_k - z_{k-1}\|^2 + 4 \left( \frac{\gamma(1-p)}{k+1} - \frac{\gamma(1-p)}{k} \right)^2 \|z_0 - z_k\|^2$$

$$\leq 4(1-p)^2 \left( \frac{\gamma^2}{k^2} + \frac{R^2}{k^{2p}} \right) \|z_k - z_{k-1}\|^2 + 4(1-p)^2 \left( \frac{p^2 R^2}{k^{2+2p}} + \frac{\gamma^2}{k^4} \right) \|z_0 - z_k\|^2$$

where the first and second inequalities follow from (6) and the third inequality follows from Assumptions (A3) and Lemma 8. □

**Main proof.** In Section 4.1, we showed

$$\frac{d}{dt}\frac{1}{2}\|\dot{z}(t)\|^2 \leq -\frac{\gamma}{t}\|\dot{z}(t)\|^2 + \frac{\gamma}{t^2}\langle z(t) - z_0, \dot{z}\rangle$$

in continuous time. We mimic analogous calculations in the discrete-time setup:

$$\frac{1}{2}\|z_{k+1} - z_k\|^2 - \frac{1}{2}\|z_k - z_{k-1}\|^2$$

$$= \left\langle \frac{1}{2}(z_{k+1} - z_{k-1}), z_{k+1} - 2z_k + z_{k-1}\right\rangle$$

$$= -\frac{1-p}{k^p}\langle z_k - z_{k-1}, g_k - g_{k-1}\rangle + (1-p)\frac{(k+1)^p - k^k}{k^p(k+1)^p}\langle z_k - z_{k-1}, g_k\rangle - \frac{\gamma(1-p)}{k}\|z_k - z_{k-1}\|^2$$

$$\quad - \frac{\gamma(1-p)}{k(k+1)}\langle z_k - z_{k-1}, z_0 - z_k\rangle + \frac{1}{2}\|z_{k+1} - 2z_k + z_{k-1}\|^2$$

$$\leq (1-p)\frac{(k+1)^p - k^k}{k^p(k+1)^p}\langle z_k - z_{k-1}, g_k\rangle - \frac{\gamma(1-p)}{k}\|z_k - z_{k-1}\|^2$$

$$\quad - \frac{\gamma(1-p)}{k(k+1)}\langle z_k - z_{k-1}, z_0 - z_k\rangle + \frac{1}{2}\|z_{k+1} - 2z_k + z_{k-1}\|^2$$

$$= -\left(\frac{\gamma(1-p)}{k} + \frac{(k+1)^p - k^p}{k^p} - \frac{\gamma(1-p)}{2k(k+1)} + \frac{\gamma(1-p)}{2}\frac{(k+1)^p - k^p}{k^p(k+1)}\right)\|z_k - z_{k-1}\|^2$$

$$\quad - \frac{(k+1)^p - k^p}{k^p}\langle z_k - z_{k-1}, z_{k+1} - 2z_k + z_{k-1}\rangle + \frac{1}{2}\|z_{k+1} - 2z_k + z_{k-1}\|^2$$

$$\quad - \gamma(1-p)\left(\frac{1}{k(k+1)} - \frac{(k+1)^p - k^p}{k^p(k+1)}\right)\left\langle z_k - z_{k-1}, z_0 - \frac{1}{2}(z_k + z_{k1})\right\rangle$$

$$\leq -\left(\frac{\gamma(1-p)}{k} + \frac{p}{k} - \frac{p(1-p)}{2k^2} - \frac{\gamma(1-p)}{2k(k+1)} + \frac{\gamma p(1-p)}{2(k+1)^2}\right)\|z_k - z_{k-1}\|^2$$

$$\quad - \frac{(k+1)^p - k^p}{k^p}\langle z_k - z_{k-1}, z_{k+1} - 2z_k + z_{k-1}\rangle + \frac{1}{2}\|z_{k+1} - 2z_k + z_{k-1}\|^2$$

$$\quad - \gamma(1-p)\left(\frac{1}{k(k+1)} - \frac{(k+1)^p - k^p}{k^p(k+1)}\right)\left\langle z_k - z_{k-1}, z_0 - \frac{1}{2}(z_k + z_{k1})\right\rangle$$

$$\leq -\left(\frac{\gamma(1-p)}{k} + \frac{p}{k} - \frac{p(1-p)}{2k^2} - \frac{\gamma(1-p)}{2k(k+1)} + \frac{\gamma p(1-p)}{2(k+1)^2}\right)\|z_k - z_{k-1}\|^2$$

$$\frac{p}{2k^2}\|z_k - z_{k-1}\|^2 + \frac{p}{2}\|z_{k+1} - 2z_k + z_{k-1}\|^2$$

$$\quad + \frac{1}{2}\|z_{k+1} - 2z_k + z_{k-1}\|^2$$

$$\quad - \gamma(1-p)\left(\frac{1}{k(k+1)} - \frac{(k+1)^p - k^p}{k^p(k+1)}\right)\left\langle z_k - z_{k-1}, z_0 - \frac{1}{2}(z_k + z_{k1})\right\rangle$$

$$= -\left(\frac{\gamma(1-p)}{k} + \frac{p}{k} - \frac{p(1-p)}{2k^2} - \frac{\gamma(1-p)}{2k(k+1)} + \frac{\gamma p(1-p)}{2(k+1)^2} - \frac{p}{2k^2}\right)\|z_k - z_{k-1}\|^2$$

$$\quad + \frac{1+p}{2}\|z_{k+1} - 2z_k + z_{k-1}\|^2$$

$$\quad - \frac{\gamma(1-p)^2}{k(k+1)}\left\langle z_k - z_{k-1}, z_0 - \frac{1}{2}(z_k + z_{k1})\right\rangle$$

$$\quad - \gamma(1-p)\underbrace{\left(\frac{1}{k(k+1)} - \frac{(k+1)^p - k^p}{k^p(k+1)} - \frac{1-p}{k(k+1)}\right)}_{=C_1(k,p)}\left\langle z_k - z_{k-1}, z_0 - \frac{1}{2}(z_k + z_{k1})\right\rangle$$

where the first inequality follows from (1), the monotonicity inequality, the second inequality follows from Lemma 8 and (6), and the third inequality follows from Lemma 8 and (5), Young's inequality, with $\varepsilon = k$.

By Lemma 10, $|C_1(k,p)| \le \frac{p(1-p)}{2k^3}$. Using (5), Young's inequality, with $\varepsilon = 1/k$ and (6) we get

$$- \gamma(1-p) \left( \frac{1}{k(k+1)} - \frac{(k+1)^p - k^p}{k^p(k+1)} - \frac{1-p}{k(k+1)} \right) \left\langle z_k - z_{k-1}, z_0 - \frac{1}{2}(z_k + z_{k1}) \right\rangle$$

$$\le \frac{\gamma p(1-p)^2}{4k^2} \|z_k - z_{k-1}\|^2 + \frac{\gamma p(1-p)^2}{4k^4} \|z_0 - \frac{1}{2}(z_k + z_{k-1})\|^2$$

$$\le \frac{\gamma p(1-p)^2}{4k^2} \|z_k - z_{k-1}\|^2 + \frac{\gamma p(1-p)^2}{8k^4} \left( \|z_0 - z_k\|^2 + \|z_0 - z_{k-1}\|^2 \right).$$

Putting these together we get

$$\frac{1}{2}\|z_{k+1} - z_k\|^2 - \frac{1}{2}\|z_k - z_{k-1}\|^2 + \frac{1}{k+1}\|z_k - z_{k-1}\|^2 - \frac{\gamma(1-p)^2}{k(k+1)} \left\langle z_k - z_{k-1}, \frac{1}{2}(z_k + z_{k1}) - z_0 \right\rangle$$

$$\le - \left( \frac{(\gamma-1)(1-p)}{k} + \frac{\gamma p(1-p)}{2(k+1)^2} - \frac{p(1-p)}{2k^2} - \frac{\gamma(1-p)}{2k(k+1)} - \frac{p}{2k^2} - \frac{\gamma p(1-p)^2}{4k^2} \right) \|z_k - z_{k-1}\|^2$$

$$+ \frac{1+p}{2}\|z_{k+1} - 2z_k + z_{k-1}\|^2 + \frac{\gamma p(1-p)^2}{8k^4} \left( \|z_0 - z_k\|^2 + \|z_0 - z_{k-1}\|^2 \right)$$

With Lemma 15 and Lemma 16, we get

$$\frac{1}{2}\|z_{k+1} - z_k\|^2 - \frac{1}{2}\|z_k - z_{k-1}\|^2 + \frac{1}{k+1}\|z_k - z_{k-1}\|^2 - \frac{\gamma(1-p)^2}{k(k+1)} \left\langle z_k - z_{k-1}, \frac{1}{2}(z_k + z_{k1}) - z_0 \right\rangle$$

$$\le - \left( \frac{(\gamma-1)(1-p)}{k} - \frac{2(1+p)(1-p)^2 R^2}{k^{2p}} + \frac{\gamma p(1-p)}{2(k+1)^2} - \frac{p(1-p)}{2k^2} - \frac{\gamma(1-p)}{2k(k+1)} - \frac{p}{2k^2} - \frac{\gamma p(1-p)^2}{4k^2} - \frac{2\gamma^2(1+p)(1-p)^2}{k^2} \right) \|z_k - z_{k-1}\|^2$$

$$+ \left( 2(1+p)(1-p)^2 \left( \frac{p^2 R^2}{k^{2+2p}} + \frac{\gamma^2}{k^4} \right) + \frac{\gamma p(1-p)^2}{8k^4} \right) C_2$$

$$\le \underbrace{\left( -\frac{(\gamma-1)(1-p)}{k} + \mathcal{O}\left( \frac{1}{k^{2p}} \right) \right)}_{= C_3(k,\gamma,p,R)} \|z_k - z_{k-1}\|^2 + \mathcal{O}\left( \frac{1}{k^{2+2p}} \right)$$

Note that there is a $K \in \mathbb{N}$ such that $C_3(k,\gamma,p,R) \le 0$ for all $k \ge K$ (with $\gamma$, $p$, and $R$ fixed).

In Section 4.1, we multiplied the established inequality by $t^2$ and integrating both sides to get

$$\frac{t^2}{2}\|\dot{z}(t)\|^2 \le \frac{\gamma}{2}\|z(t) - z_0\|^2.$$

We mimic analogous calculations in the discrete-time setup. Multiply both sides with $k(k+1)$ and sum both sides from $k=1$ to $k=k$, and apply Lemma 11 and Lemma 12 to get

$$\frac{k(k+1)}{2}\|z_{k+1} - z_k\|^2 \le \frac{\gamma(1-p)^2}{2}\|z_k - z_0\|^2 + C_4 + \mathcal{O}\left( \frac{1}{k^{2p-1}} \right)$$

where $C_4 < \infty$ since $C_3(k,\gamma,p,R) > 0$ for only finitely many $k$. Reorganizing we get

$$\frac{k(k+1)(1-p)^2}{2(k+1)^{2p}}\|g_k\|^2 + \frac{k(1-p)^2\gamma^2}{2(k+1)}\|z_0 - z_k\|^2 - \frac{k(1-p)^2\gamma}{(k+1)^p}\langle g_k, z_0 - z_k \rangle$$

$$\le \frac{\gamma(1-p)^2}{2}\|z_k - z_0\|^2 + C_4 + \mathcal{O}\left( \frac{1}{k^{2p-1}} \right)$$

Reorganizing yet again we get

$$\frac{k(k+1)(1-p)^2}{2(k+1)^{2p}}\|g_k\|^2 - \frac{k(1-p)^2\gamma}{(k+1)^p}\langle g_k, z_0 - z_k \rangle$$

$$\le \frac{\gamma(1-p)^2}{2}\left( 1 - \frac{\gamma k}{k+1} \right)\|z_k - z_0\|^2 + C_4 + \mathcal{O}\left( \frac{1}{k^{2p-1}} \right)$$

$$\le C_4 + \mathcal{O}\left( \frac{1}{k^{2p-1}} \right),$$

where we use the assumption that $\gamma \geq 2$. Reorganizing again, we get

$$
\begin{aligned}
\|g_k\|^2 &\leq \frac{2\gamma}{(k+1)^{1-p}}\langle g_k, z_0 - z_k\rangle + \frac{2C_4}{(1-p)^2 k(k+1)^{1-2p}} + \mathcal{O}\left(\frac{1}{k}\right) \\
&\leq \frac{2\gamma}{(k+1)^{1-p}}\langle g_k, z_0 - z_\star\rangle + \frac{4C_4}{(1-p)^2(k+1)^{2-2p}} + \mathcal{O}\left(\frac{1}{k}\right) \\
&\leq \frac{1}{2}\|g_k\|^2 + \frac{2\gamma^2}{(k+1)^{2-2p}}\|z_0 - z_\star\|^2 + \frac{4C_4}{(1-p)^2(k+1)^{2-2p}} + \mathcal{O}\left(\frac{1}{k}\right)
\end{aligned}
$$

for $k \geq 1$, where the second inequality follows from (1), the monotonicity inequality, and the third inequality follows from (5), Young's inquality, with $\varepsilon = \gamma/(k+1)^{1-p}$. Finally, we have

$$
\|g_k\|^2 \leq \frac{C}{k^{2-2p}} + \mathcal{O}\left(\frac{1}{k}\right)
$$

with $C = 4\gamma^2 + 8C_4/(1-p)^2$. $\qquad\square$

## F  ANALYSIS OF THEOREM 4

For convenience, we restate the update, assumptions, and the theorem:

$$
z_{k+1} = z_k - \frac{1-p}{(k+1)^p}g(z_k; \omega_k) + \frac{(1-p)\gamma}{(k+1)^{1-\varepsilon}}(z_0 - z_k) \tag{SSSGD-A}
$$

$$
L \text{ is convex-concave and has a saddle point} \tag{A0}
$$
$$
\mathbb{E}_{\omega_1, \omega_2}\|g(z_1; \omega_1) - g(z_2; \omega_2)\|^2 \leq R_1^2\|z_1 - z_2\|^2 + R_2^2 \quad \forall\, z_1, z_2 \in \mathbb{R}^{m+n} \tag{A2}
$$

**Theorem 4.**   *Assume* (A0) *and* (A2). *If* $p \in (1/2, 1)$, $\varepsilon \in (0, 1/2)$, *and* $\gamma > 0$, *then SSSGD-A converges in the sense of* $z_k \xrightarrow{L^2} z_\star$, *where* $z_\star$ *is a saddle point.*

To clarify, we do not assume $L$ is differentiable for Theorem 4.

**Proof outline.**   The key insight is to define $\zeta_k$ to be something like a "fixed point" of the $k$-th iteration of SSSGD-A and then to show $z_k$ shrinks towards to $\zeta_k$ in the following sense

$$
\|z_{k+1} - \zeta_{k+1}\|^2 \leq (1 - \text{something})\|z_k - \zeta_k\|^2 + (\text{something small}).
$$

Lemma 17 states that $\zeta_k$ slowly (stably) converges to a solution. Using the fact that $z_k$ shrinks towards $\zeta_k$ and the fact that $\zeta_k$ is a slowly moving target converging to a solution, we conclude $z_k$ converges to a solution.

**Preliminary definition and result.**   More precisely, we define $\zeta_k$ to satisfy

$$
\zeta_k \in \zeta_k - \frac{1-p}{(k+1)^p}G(\zeta_k) + \frac{(1-p)\gamma}{(k+1)^{1-\varepsilon}}(z_0 - \zeta_k).
$$

(However, $\zeta_k$ is not actually a fixed point, since SSSGD-A has noise and since $G$ is a multi-valued operator.) We equivalently write

$$
\zeta_{k+1} = \left(I + \frac{(k+1)^{1-p-\varepsilon}}{\gamma}G\right)^{-1}(z_0).
$$

**Lemma 17** (Proposition 23.31 and Theorem 23.44 of Bauschke & Combettes (2017)). *Let $G$ be a maximal monotone operator such that* $\mathrm{Zer}(G) \neq \emptyset$. *Then* $(I + \tau G)^{-1}(z_0) \to P_{\mathrm{Zer}(G)}(z_0)$ *and*

$$
\|(I + (\tau + s)G)^{-1}(z_0) - (I + \tau G)^{-1}(z_0)\| \leq \mathcal{O}\left(\frac{s}{\tau}\right)
$$

*for any* $s \geq 0$ *as* $\tau \to \infty$.

### F.1 Proof of Theorem 4

**Main proof.** Since $1 - p - \varepsilon > 0$, Lemma 17 gives us

$$\zeta_k \to P_{\mathrm{Zer}(G)}(z_0).$$

Then we have

$$\mathbb{E}\left[\|z_{k+1} - \zeta_{k+1}\|^2 \mid \mathcal{F}_k\right]$$

$$= \mathbb{E}\left[\left\|z_k - \zeta_k - \frac{1-p}{(k+1)^p}g(z_k; \omega_k) + \frac{(1-p)\gamma}{(k+1)^{1-\varepsilon}}(z_0 - z_k) + \zeta_k - \zeta_{k+1}\right\|^2 \mid \mathcal{F}_k\right]$$

$$= \|z_k - \zeta_k\|^2 - \left\langle \frac{1-p}{(k+1)^p}G(z_k) + \frac{(1-p)\gamma}{(k+1)^{1-\varepsilon}}(z_k - z_0), z_k - \zeta_k \right\rangle$$

$$\quad + \langle z_k - \zeta_k, \zeta_k - \zeta_{k+1}\rangle$$

$$\quad + \mathbb{E}\left[\left\|\frac{1-p}{(k+1)^p}g(z_k; \omega_k) - \frac{(1-p)\gamma}{(k+1)^{1-\varepsilon}}(z_0 - z_k) - \zeta_k + \zeta_{k+1}\right\|^2 \mid \mathcal{F}_k\right]$$

$$\leq \left(1 - \frac{(1-p)\gamma}{(k+1)^{1-\varepsilon}}\right)\|z_k - \zeta_k\|^2 + \|z_k - \zeta_k\|\,\|\zeta_k - \zeta_{k+1}\|$$

$$\quad + \mathbb{E}\left[\mathcal{O}\left(\frac{1}{(k+1)^{2p}}\right)\|g(z_k; \omega_k)\|^2 \mid \mathcal{F}_k\right] + \mathcal{O}\left(\frac{1}{(k+1)^{2(1-\varepsilon)}}\right)\|z_0 - z_k\|^2 + \mathcal{O}\left(\frac{1}{(k+1)^2}\right)$$

$$\leq \left(1 - \frac{(1-p)\gamma}{(k+1)^{1-\varepsilon}}\right)\|z_k - \zeta_k\|^2 + \mathcal{O}(1/k)\|z_k - \zeta_k\|$$

$$\quad + \mathcal{O}\left(\frac{1}{(k+1)^{2p}}\right)(R_3^2\|z_0 - z_k\|^2 + R_4^2) + \mathcal{O}\left(\frac{1}{(k+1)^{2(1-\varepsilon)}}\right)\|z_0 - z_k\|^2 + \mathcal{O}\left(\frac{1}{(k+1)^2}\right),$$

where the first inequality follows from (1), the monotonicity inequality, Cauchy-Schwartz inequality, and (6), Now we take the full expectation to get

$$\mathbb{E}\left[\|z_{k+1} - \zeta_{k+1}\|^2\right]$$

$$\leq \left(1 - \mathcal{O}\left(\frac{1}{(k+1)^{1-\varepsilon}}\right) + \mathcal{O}\left(\frac{1}{(k+1)^{2p}}\right) + \mathcal{O}\left(\frac{1}{(k+1)^{2(1-\varepsilon)}}\right)\right)\mathbb{E}\left[\|z_k - \zeta_k\|^2\right]$$

$$\quad + \mathcal{O}(1/k)\mathbb{E}\left[\|z_k - \zeta_k\|^2\right]^{1/2}$$

$$\quad + \mathcal{O}\left(\frac{1}{(k+1)^{2p}}\right)(\|z_0 - z_\star\|^2 + 1) + \mathcal{O}\left(\frac{1}{(k+1)^{2(1-\varepsilon)}}\right)\|z_0 - z_\star\|^2 + \mathcal{O}\left(\frac{1}{(k+1)^2}\right),$$

where we used $\mathbb{E}[\|z_k - \zeta_k\|]^2 \leq \mathbb{E}[\|z_k - \zeta_k\|^2]$. Applying Lemma 14, we get $\mathbb{E}\left[\|z_k - \zeta_k\|^2\right] \to 0$. Since $\zeta_k \to P_{\mathrm{Zer}(G)}(z_0)$, we conclude $z_k \xrightarrow{L^2} P_{\mathrm{Zer}(G)}(z_0)$. □

## G Experiment details

In this section, we prodvide further details of the experiments of Section 5. Our Optimistic Adam is a variation of the Optimistic Adam of (Daskalakis et al., 2018), which uses $\beta = 1$ while we allow for a general optimism rate $\beta > 0$. For Anchored Adam, we do not diminish the strength of the anchor proportional to $1/k^{1-\varepsilon}$ since Adam does not diminish the learning rate. Rather, we maintain a constant anchor strength $\gamma$ but refresh the anchor point every $T$ iterations. The notation $\nabla^2$ in algorithm tables denote the element-wise square operation.

| Generator |
| --- |
| latent space 100 (Gaussian noise) |
| dense 128 lReLU |
| dense 256 batchnorm lReLU |
| dense 512 batchnorm lReLU |
| dense 1024 batchnorm lReLU |
| dense 1024 tanh |
| Discriminator |
| Resize the input image $28 \times 28$ to $32 \times 32$ |
| dense 512 lReLU |
| dense 256 lReLU |
| dense 1 |

Table 1: Generator and discriminator architectures for the MNIST experiment

---

**Optimistic Adam**

---

**Parameters**: learning rate $\eta$, exponential decay rates for moment estimates $\beta_1$, $\beta_2 \in [0, 1)$, optimism rate $\rho > 0$, and initial parameters $z_0$
**Repeat** $k = 0, 1, 2 \ldots, K$ (iteration):
$\quad$ Compute stochastic gradient $\nabla_{z,k} = G(z_k)$
$\quad$ Update biased estimate of first moment: $m_k = \beta_1 m_{k-1} + (1 - \beta_1)\nabla_{z,k}$
$\quad$ Update biased estimate of second moment: $v_k = \beta_2 v_{k-1} + (1 - \beta_2)\nabla_{z,k}^2$
$\quad$ Scale the step-size: $\hat{\eta}_k = \eta\sqrt{1 - \beta_2^k}/(1 - \beta_1^k)$
$\quad$ Perform optimistic gradient step: $z_k = z_{k-1} - \hat{\eta}_k(1 + \rho)\frac{m_k}{\sqrt{v_k}+\epsilon} + \hat{\eta}_{k-1}\rho\frac{m_{k-1}}{\sqrt{v_{k-1}}+\epsilon}$
**Return** $z_K$

---

**Anchored Adam**

---

**Parameters**: learning rate $\eta$, exponential decay rates for moment estimates $\beta_1$, $\beta_2 \in [0, 1)$, anchor rate $\gamma > 0$, anchor update period $T$, and initial parameters $z_0$
**Repeat** $k = 0, 1, 2 \ldots, K$ (iteration):
$\quad$ set anchor $a_k = z_k$ if $\mod(k, T) = 0$ else $a_k = a_{k-1}$
$\quad$ Compute stochastic gradient $\nabla_{z,k} = G(z_k)$
$\quad$ Update biased estimate of first moment: $m_k = \beta_1 m_{k-1} + (1 - \beta_1)\nabla_{z,k}$
$\quad$ Update biased estimate of second moment: $v_k = \beta_2 v_{k-1} + (1 - \beta_2)\nabla_{z,k}^2$
$\quad$ Scale the step-size: $\hat{\eta}_k = \eta\sqrt{1 - \beta_2^k}/(1 - \beta_1^k)$
$\quad$ Perform anchored gradient step: $z_k = z_{k-1} - \hat{\eta}_k\frac{m_k}{\sqrt{v_k}+\epsilon} + \gamma(a_k - z_{k-1})$
**Return** $z_K$

---

| |
|---|
| batch size = 64 |
| Adam learning rate = 0.0002 |
| Adam $\beta_1 = 0.5$ |
| Adam $\beta_2 = 0.999$ |
| max iteration = 200000 |
| GAN objctive = "WGAN-GP" |
| Gradient penalty parameter $\lambda = 10$ |
| $n_{\mathrm{dis}} = 5$ |
| Optimizer = "Adam", "Optimistic Adam", or "Anchored Adam" |
| Optimism rate $\rho = 1$ |
| Anchor rate $\gamma = 1$ |
| Anchor refresh period $T = 10000$ |

Table 2: Hyperparameters for the MNIST experiment

| Generator |
|---|
| latent space 128 (Gaussian noise) |
| dense $4 \times 4 \times 512$ batchnorm ReLU |
| $4 \times 4$ conv.T stride=2 256 batchnorm ReLU |
| $4 \times 4$ conv.T stride=2 128 batchnorm ReLU |
| $4 \times 4$ conv.T stride=2 64 batchnorm ReLU |
| $4 \times 4$ conv.T stride=1 3 weightnorm tanh |
| **Discriminator** |
| Input Image $32 \times 32 \times 3$ |
| $3 \times 3$ conv. stride=1 64 lReLU |
| $3 \times 3$ conv. stride=2 128 lReLU |
| 3 conv. stride=1 128 lReLU |
| 3 conv. stride=2 256 lReLU |
| 3 conv. stride=1 256 lReLU |
| 3 conv. stride=2 512 lReLU |
| 3 conv. stride=1 512 lReLU |
| dense 1 |

Table 3: Generator and discriminator architectures for the CIFAR-10 experiment

| |
|---|
| batch size = 64 |
| Adam learning rate = 0.0001 |
| Adam $\beta_1 = 0.0$ |
| Adam $\beta_2 = 0.9$ |
| max iteration = 100000 |
| GAN objctive = "WGAN-GP" |
| Gradient penalty parameter $\lambda = 1$ |
| $n_{\mathrm{dis}} = 1$ |
| Optimizer = "Adam", "Optimistic Adam", or "Anchored Adam" |
| Optimism rate $\rho = 1$ |
| Anchor rate $\gamma = 1$ |
| Anchor refresh period $T = 10000$ |

Table 4: Hyperparameters for the CIFAR-10 experiment

