# OpenReview forum: "ODE Analysis of Stochastic Gradient Methods with Optimism and Anchoring  for Minimax Problems and GANs"
_ICLR.cc/2020/Conference — Reject_

### Official Review · AnonReviewer2 · 2019-10-21
**Official Blind Review #2**

**Rating:** 1

**Review:**

The paper studies last-iterate convergence of simultaneous gradient descent and related algorithms in (convex-concave) GANs.

The experiments are very weak. Figure 4 shows that anchored Adam outperforms Adam and optimistic Adam in terms of FID on MNIST, but not CIFAR-10.

The authors cite many of the vast number of algorithms that have been proposed to train GANs (Heusel, Mescheder, Gidel, Gemp, and on and on ... and on…) and discuss some of them in the analysis. However, when it comes to experiments, they only compare against vanilla Adam (!) and Optimism. I would summarize the experiments as *suggesting* that anchoring doesn’t hurt on MNIST or CIFAR-10. The paper doesn’t tune beta and gamma, so it’s hard to be sure.

The motivation for the paper is GANs, but GANs are not convex-concave. The analysis is therefore not directly relevant. From the experiments, it is not clear *at all* whether anchored Adam is *actually* an improvement in practice over any of the alternative algorithms that the authors discuss or cite.

In short, the contribution is unclear. The analysis is better suited to a venue like COLT. To be a reasonable ICLR submission, the paper needs to compare against more baselines at a bare minimum.


**Experience Assessment:**

I have published one or two papers in this area.

**Review Assessment: Checking Correctness Of Derivations And Theory:**

I assessed the sensibility of the derivations and theory.

**Review Assessment: Checking Correctness Of Experiments:**

I carefully checked the experiments.

**Review Assessment: Thoroughness In Paper Reading:**

I read the paper thoroughly.

---

> ### Author Response · Authors · 2019-11-15
> **Theory papers do not belong in ICLR?**
>
> We agree with Reviewer #2 that the main contribution of this work is the theory, while the experiments are at best suggestive. (Reviewer #2 offers COLT, the more theoretical venue, as a more appropriate venue.)
>
> We firmly disagree with the statement that our theoretical contribution is not suited for ICLR. We cite several ICLR papers with similar theoretical contributions and simple experiments:
> - Daskalakis et al., ICLR, 2018.
> - Gidel et al., ICLR, 2019.
> - Mertikopoulos et al., ICLR, 2019.
> We decided to submit to ICLR as our paper continues this line of work by offering stronger theoretical results. If the reviewers believe the experiments are a distraction, we are happy to remove them.
>
> We ask the following question to Reviewer #2: Are theory papers without a significant experimental component welcome at ICLR?
> We would appreciate a clear statement, even if we disagree. Clarifying the basis of the decision on the public forum will be of value to the ML community, as it will serve as a guideline for future authors in determining whether ICLR is an appropriate venue for their theoretical work.
>
> Finally, please see our other comment on the convex-concavity assumption.

---

### Official Review · AnonReviewer1 · 2019-10-23
**Official Blind Review #1**

**Rating:** 6

**Review:**

*Summary*
This paper provides the analysis of three algorithms in the context of minmax convex concave games: Simultaneous stochastic subgradient method, Simultaneous gradient with optimism and Simultaneous gradient with anchoring. These three algorithm are first analysed in continuous time with an ODE perspective and then leverage these intuitions and techniques to analyze the discrete time versions.

I think that these contributions are of interest of ICLR community, however, I have some concerns regarding the presentations of the results.

*Decision*
I vote for a weak accept that could move to an accept if the authors improve the presentation of the paper:
The paragraph “Regularized dynamics and convergence’ in Section 3.1 in quite hard to follow. This subsection is basically the proof of the convergence of the continuous version of  GD-O. Stating the result before the proof would help the reader to understand where the authors want to go.
Very same point for The subsection 4.1
For the stochastic version of SimGD-A a new parameter $\epsilon$ is introduced without any comment or description on why it is necessary.
A FID above 40 for MNIST is very far from standard results (that are below 1). Thus I am not convinced by the practical advantage of Anchored Adam on these models that have performances results very far from the standard ones.  (for instance between 20 and 25 for CIFAR10 is very reasonable). It is maybe because you do not use convolutional layers in your architecture. I think that using a DGAN architecture (for instance the one from the pytorch tutorial https://pytorch.org/tutorials/beginner/dcgan_faces_tutorial.html) would give the expected results (i.e. FID close to 0).

*Questions*

- When you cite Lassale Principle you mention that if $z_\infty$ is a limit point of $z(t)$ then starting at $z_\infty$, you stay at a constant distance to $z_*$. But in the bilinear example you give any point in the circle is not a limit point (size no dynamics converge to it). I guess when you said limit point you meant adherent point ?
- Is the condition $\epsilon$ only necessary for the proof ? Does $\epsilon =1$ work in practice ?
If no, what is the best value for $\epsilon$ ?


**Experience Assessment:**

I have published in this field for several years.

**Review Assessment: Checking Correctness Of Derivations And Theory:**

I assessed the sensibility of the derivations and theory.

**Review Assessment: Checking Correctness Of Experiments:**

I carefully checked the experiments.

**Review Assessment: Thoroughness In Paper Reading:**

I read the paper thoroughly.

---

> ### Author Response · Authors · 2019-11-15
> **Thank you for the thoughtful comments**
>
> We thank Reviewer #1 for the thoughtful feedback. In our revision, we have followed Reviewer #1's suggestions to improve the presentation of the paper. We respond to the individual comments below.
>
> "[Section 3.1 and 4.1 are] basically the proof of the convergence ... Stating the result before the proof would help the reader to understand where the authors want to go."
>
> Thank you for the suggestion. We have updated the structure to improve the readability. We avoided saying "Theorem" or "convergence proof"  to avoid discussing the existence and differentiability of solutions to the continuous-time ODEs. (We do not see such rigorous treatment of the continuous-time setup to be important since the continuous-time analysis serves merely as an illustration and inspiration for the discrete-time algorithm.)
>
>
> "For the stochastic version of SimGD-A a new parameter is introduced without any comment or description on why it is necessary. Question) Is the condition only necessary for the proof? Does work in practice? If no, what is the best value for?"
>
> Due to page limitations, we deferred the discussion on this matter to the appendix. The proof requires $\varepsilon>0$, but we believe this is an artifact of the proof. In particular, we conjecture that Lemma 17 holds with $o(s/\tau)$ rather than $\mathcal{O}(s/\tau)$, and, if so, it is possible to establish convergence with $\varepsilon=0$. In practice, we observed that $\varepsilon=0$ to work well.
>
> "A FID above 40 for MNIST is very far from standard results."
>
> Indeed, our experiments are far from the state-of-the-art since it is a simple architecture without any modern techniques (e.g. spectral normalization). Our experiments only demonstrate that anchoring helps in some setups, not necessarily state-of-the-art setups. If the reviewers believe the experiments are a distraction, we are happy to remove them. We ask the reviewers to consider the theoretical contribution in their decision.
>
> "Question) When you cite Lassale Principle you mention that if $z_\infty$ is a limit point of $z(t)$ then starting at $z_\infty$, you stay at a constant distance to $z_*$. But in the bilinear example you give any point in the circle is not a limit point (size no dynamics converge to it). I guess when you said limit point you meant adherent point?"
>
> Thank you for pointing this out. Indeed, "limit point" is not the correct word. We should have said "adherent point" or, since we are talking about sequences,  "cluster point" as defined in
> https://en.wikipedia.org/wiki/Limit_point#For_sequences_and_nets

---

### Official Review · AnonReviewer3 · 2019-10-24
**Official Blind Review #3**

**Rating:** 6

**Review:**

This paper analyzes the dynamics of stochastic gradient descent when applied to convex-concave games (motivated by the game used to train GANs, which is typically not convex-concave), as well as the previously proposed GD with optimism and a new anchored GD algorithm that provably converges under weaker assumptions than SGD or SGD with optimism.

This seems like a nice contribution to the GAN theory literature, and the anchored SSSGD algorithm might well have significant practical value (although more thorough experiments would be needed to make this claim). As such I recommend acceptance.

The obvious critique to raise of this kind of work is that the GAN problems that motivate it are clearly not convex-concave, and so it is unclear how or whether the results can or should inform practice. The simplest way to make the case for that kind of relevance is empirically, so I'd recommend that the authors consider what kinds of GAN experiments would support (or falsify) the claim that their theoretical results have some hope of generalizing to the non-convex-concave setting. Figure 4 is suggestive, but it doesn't say much except that anchored Adam might occasionally be a good choice (and sometimes isn't). The samples in Figure 3 are pretty far from the state of the art, and in any case Figure 3 doesn't even say which training method generated them.

Finally, PLEASE don't include acknowledgments in a paper that's under double-blind review; it compromises your anonymity, which in principle could be grounds for a desk rejection.

**Experience Assessment:**

I have read many papers in this area.

**Review Assessment: Checking Correctness Of Derivations And Theory:**

I assessed the sensibility of the derivations and theory.

**Review Assessment: Checking Correctness Of Experiments:**

I carefully checked the experiments.

**Review Assessment: Thoroughness In Paper Reading:**

I read the paper at least twice and used my best judgement in assessing the paper.

---

> ### Author Response · Authors · 2019-11-15
> **Thank you for the thoughtful comments**
>
> We thank Reviewer #3 for the thoughtful feedback.
>
> Please see our other comment on the convex-concavity assumption.
>
> We agree the experiments are at best suggestive, and far from the state-of-the-art since it is a simple architecture without any modern techniques, and we appreciate that Reviewer #3 is recognizing that the main contribution of this paper is theoretical. If the reviewers believe the experiments are a distraction, we are happy to remove them. We ask the reviewers to consider the theoretical contribution in their decision.
>
> In the revision, we have clarified which training method generated Figure 3.
>
> We will make sure to exclude acknowledgments in future double-blind submissions. We apologize for the mistake.

---

### Author Response · Authors · 2019-11-15
**Convex-concavity assumption is, comparatively, not unreasonable**

This comment is in response to comments from Reviewers #2 and #3.

In our analysis, we assume convex-concavity, even though GANs are not convex-concave. However, prior theoretical papers make assumptions that are equally or further unrealistic, as we discuss in the introduction. Common assumptions are:
 - GAN is quadratic in the discriminator and generator parameters.
 - Use full gradient, not stochastic gradients.
(Examples of recent published work under such assumptions include: Mescheder et al., ICML, 2019; Daskalakis and Panageas, NeurIPS, 2018; Gidel et al., AISTATS, 2019.) Compared to these assumptions, convex-concavity is no more unrealistic. We believe the analyses under the different (unrealistic) assumptions provide complementary insights into the training dynamics of GANs.

In the past, the analysis of SGD for minimization in ML initially focused on the convex case. The highly cited AdaGrad paper by Duchi et al. in 2011 is one such example
http://jmlr.org/papers/v12/duchi11a.html
As our understanding of SGD matured, attention shifted towards more realistic non-convex ML minimization setups. We believe the analysis of SGD-type methods for minimax problems will take a similar course. Currently, the minimax theory relies on unrealistic assumptions, but once our understanding matures, the field will be able to move on to more realistic setups.

We believe the validity of the convex-concavity assumption should be judged in comparison to the prior theoretical works, rather than how well the assumption matches the empirical practice. Our assumption of convex-concave (but non-differentiable and stochastic gradients) is very reasonable compared to the assumptions used in prior theoretical works.

---

### Decision · Program_Chairs · 2019-12-19

**Decision:**

Reject

**Comment:**

Motivated by GANs, the authors study the convergence of stochastic subgradient
descent on convex-concave minimax games.
They introduced an improved "anchored" SGD variant, that provably converges
under milder assumptions that the base algorithm.
It is applied to training GANs on MNIST and CIFAR-10, partially showing
improvements over alternative training methods.

A main point of criticism that the reviewers identify is the strength of the
assumptions needed for the analysis.
Furthermore, the experimental results were deemed weak as the reported scores
are far away from the SOTA, and only simple baselines were compared against.